# RAG4GFM: Bridging Knowledge Gaps in Graph Foundation Models through Graph Retrieval Augmented Generation

**Xingliang Wang**[1]     **Zemin Liu**[1,*]    **Junxiao Han**[2,*]    **Shuiguang Deng**[1]

[1]College of Computer Science and Technology, Zhejiang University, Hangzhou, China
[2]School of Computer and Computing Science, Hangzhou City University, Hangzhou, China
{wangxingliang, liu.zemin, dengsg}@zju.edu.cn, hanjx@hzcu.edu.cn

## Abstract

Graph Foundation Models (GFMs) have demonstrated remarkable potential across graph learning tasks but face significant challenges in knowledge updating and reasoning faithfulness. To address these issues, we introduce the Retrieval-Augmented Generation (RAG) paradigm for GFMs, which leverages graph knowledge retrieval. We propose RAG4GFM , an end-to-end framework that seamlessly integrates multi-level graph indexing, task-aware retrieval, and graph fusion enhancement. RAG4GFM implements a hierarchical graph indexing architecture, enabling multi-granular graph indexing while achieving efficient logarithmic-time retrieval. The task-aware retriever implements adaptive retrieval strategies for node, edge, and graph-level tasks to surface structurally and semantically relevant evidence. The graph fusion enhancement module fuses retrieved graph features with query features and augments the topology with sparse adjacency links that preserve structural and semantic proximity, yielding a fused graph for GFM inference. Extensive experiments conducted across diverse GFM applications demonstrate that RAG4GFM significantly enhances both the efficiency of knowledge updating and reasoning faithfulness[2].

## 1 Introduction

Graph representation learning [1, 2, 3] has achieved remarkable progress across diverse graph tasks, such as node classification and link prediction. Concurrently, the substantial success of Large Language Models (LLMs) [4, 5] has revolutionized natural language processing (NLP) and motivated the development of Graph Foundation Models (GFMs) [3, 6, 7]. GFMs adapt large-scale pre-training techniques to graph data, enabling powerful cross-domain generalization and multi-task adaptability for diverse graph-based applications. However, two practical challenges remain prominent: knowledge updating [8, 9, 10]i.e., keeping models current as graphs evolveand faithful reasoning [11, 12, 13]i.e., generating accurate and factually consistent outputs.

On the one hand, as graph data evolves rapidly, GFMs usually require knowledge updates within hours or a day [14, 15], massive parameter scales impose substantial computational demands [16, 17]. On the other hand, handling complex graph structures [18] and coupling GFMs with LLMs may induce hallucinations [19, 20, 21], e.g., generating fictitious features or nodes, leading to unfaithful or factually inconsistent outputs.

Some recent studies have attempted to address these challenges. Parameter-efficient fine-tuning (PEFT) methods on GFMs, such as GraphLoRA [22, 23] and G-Adapter [24], aim to reduce the cost

---

[*]Corresponding authors.
[2]Code: `https://github.com/Matrixmax/RAG4GFM`.

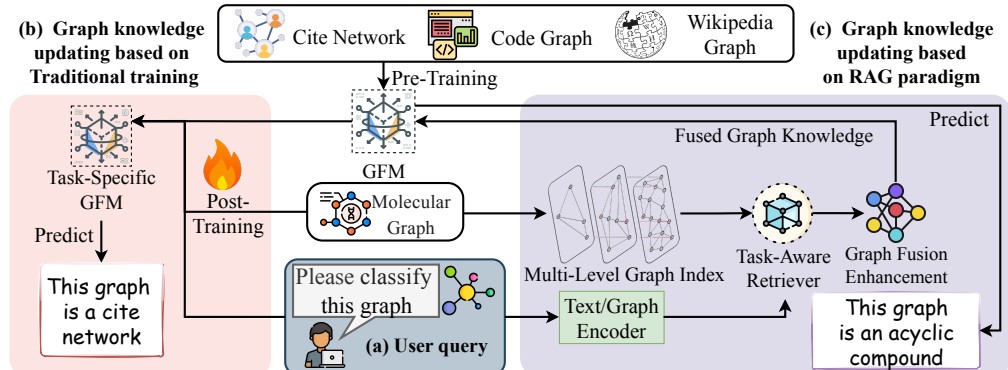

Figure 1: Comparison between conventional training-driven GFMs (b) and RAG4GFM (c). (a) A user query comprising a natural-language description and a graph; (b) Conventional GFMs rely on task-specific post-training to integrate new knowledge, incurring substantial computational cost; (c) RAG4GFM integrates external graph knowledge at inference time via retrieval and fusion, ensuring faithful reasoning without additional post-training.

of knowledge updating by adapting pre-trained models via task-specific post-training (as depicted in Figure 1(b)). However, these methods still require substantial computational resources and remain vulnerable to catastrophic forgetting. Concurrent efforts to improve reasoning faithfulness focus on enhancing the quality of training data [25, 26] or refining GFM architectures [27, 28]. Despite these advancements, reliable reasoning remains challenging because these methods largely operate within fixed parameters and training data, lacking mechanisms to dynamically ground predictions in verifiable, task-relevant graph evidence.

Retrieval-augmented generation (RAG) [29, 30] offers an appealing alternative: by retrieving external evidence at inference time, RAG circumvents frequent parameter updates, adapts to evolving corpora, and can mitigate hallucinations [31, 32] However, extending this RAG paradigm to graph data and GFMs raises three challenges: (1) *Indexing: how to build graph indices that preserve structure and serve both RAG and GFMs?* (2) *Retrieval: how to design retrieval mechanisms that accommodate task heterogeneity?* (3) *Augmentation: how to augment the task-specific query with retrieved graph evidence?*

To overcome these challenges, we propose RAG4GFM , a unified RAG framework designed explicitly for graph data, graph tasks, and GFMs. To our knowledge, it is among the first comprehensive designs that operationalize RAG for GFMs. As illustrated in Fig. 1(c), RAG4GFM augments a GFM with external graph knowledge through a multi-stage pipeline. Firstly, the "multi-level graph index" module processes raw graph data into an efficient, structure-aware index. Secondly, the "task-aware retriever" module identifies the user intent and retrieves relevant candidate subgraphs from the constructed index. Finally, the "graph fusion enhancement" module integrates retrieved subgraph knowledge into the user query, enriching its features and structure. By grounding GFM's predictions in retrieved graph evidence-rather than solely in its internal parametersRAG4GFM flexibly adapts to evolving knowledge and markedly enhances reasoning fidelity.

Specifically, we first establish an efficient and flexible multi-level graph indexing module. This module is designed to preserve graph semantics and structural topology by integrating four complementary indices: text features extracted via LM encoders, node embeddings that capture structural information via Laplacian positional encoding [33], edge-level representations, and graph-level embeddings. By leveraging the hierarchical structure index, the module achieves comprehensive semantic-structural encoding with logarithmic-time complexity.

Second, building on this index, we propose a task-aware retriever module that adapts to diverse downstream task types. This module dynamically selects appropriate indices and retrieval strategies based on the task typology (node, edge, or graph), retrieving relevant textual and structural features for node tasks, edge-level representations for edge tasks, and graph-level embeddings for graph tasks. Additionally, we use a fusion reranker to consolidate and prioritize retrieval results from both graph features and semantic spaces.

Finally, we introduce a graph fusion enhancement module that integrates retrieved evidence with the query graph effectively. This module consists of two components: a feature-fusion component

that integrates the retrieved graph features with query features based on attention weights calculated from similarity scores, while a topological-structure enhancement component augments the query graph's connectivity by combining adjacency information from relevant retrieved graphs through sparse matrix operations. This structured fusion approach surpasses sequential concatenation in traditional RAG by better aligning with graph connectivity, preserving topological information, and accommodating multimodal user queries within the GFM context.

The GFM then performs inference on this fused graph, allowing it to leverage the augmented context for more accurate predictions and a significant reduction in hallucinations. Extensive experiments across multiple GFM applications demonstrate RAG4GFM 's superiority in efficiency and reliability, with safeguards intended to minimize the risk of pre-training data contamination.

## 2   Related Work

**Graph Foundation Models.** GFMs leverage large-scale pre-training for versatile knowledge transfer, exhibiting reasoning and domain adaptation capabilities [34]. They encompass: (1) Self-supervised learning approaches, such as masked auto-encoding (e.g., GraphMAE [27]); (2) Graph-language model alignment methods that facilitate multimodal pre-training, such as GraphGPT [16]; and (3) Recent architectural innovations aimed at enhancing transferability via Mixture of Experts (MoE), such as AnyGraph [17], structural understanding through topology-aware tokenization as in Open-Graph [35], or improving generalization through property-driven training, such as GraphProp [28]. However, GFMs are constrained by inefficient knowledge updating, typically requiring extensive retraining [16], and difficulties in ensuring reasoning reliability over complex graph structures, potentially leading to biases [11].

**Graph Indexing and Retrieval.** Graph indexing and retrieval have progressed from general vector methods to structure-aware techniques. Initial advancements focused on optimizing vector-space retrieval efficiency, including adaptive algorithms, such as FLANN [36], and hierarchical graph structures for efficient search, such as HNSW [37, 38]. These formed the basis for scalable libraries such as FAISS [39]. While traditional Information Retrieval (IR) methods like BM25 [40] exist, vector-based strategies are more directly applicable to graph data retrieval. Despite these advancements, developing retrieval systems that effectively integrate multimodal data (e.g., text with graph structures) and generalize across diverse graph-specific tasks remains a significant challenge.

**Graph Retrieval-Augmented Generation.** RAG [30] enhances LLMs by integrating external knowledge through a "retrieve-generate" pipeline. Its progression includes: (1) Foundational end-to-end trainable RAG frameworks for knowledge-intensive NLP [30]; (2) Architectural refinements, including multi-hop retrieval for comprehensive knowledge gathering and integrated retriever-generator optimization [41, 42]; and (3) Recent graph-centric extensions, featuring methods that emphasize structure-aware retrieval, such as GraphRAG [43], computational optimization, such as LightRAG [44], or hybrid knowledge integration, such as HybridRAG [45]. Nevertheless, applying RAG to graph data still encounters critical challenges. Standard vector retrieval techniques often fail to adequately represent complex graph topology [46]. Moreover, the creation of specialized graph indexing and retrieval mechanisms tailored for the unique requirements of graph RAG remains an active research direction.

## 3   RAG4GFM

RAG4GFM is a RAG framework tailored for graph data and GFMs, integrating external graph knowledge into the GFM inference process. The pipeline initiates with multi-level graph indexing (Figure 2(a)), where RAG4GFM constructs HNSW-based [47] multimodal indices for graph corpora. Next, the task-aware retrieval mechanism (Figure 2(b)) processes user queries (text and graph structure) and adaptively retrieves relevant subgraphs. Subsequently, the graph fusion enhancement module (Figure 2(c)) integrates these retrieved subgraphs with the original query graph through a two-step process: attention-based feature fusion and adjacency matrix-based topological fusion. Finally, the fused graph is input to the base GFM for inference.

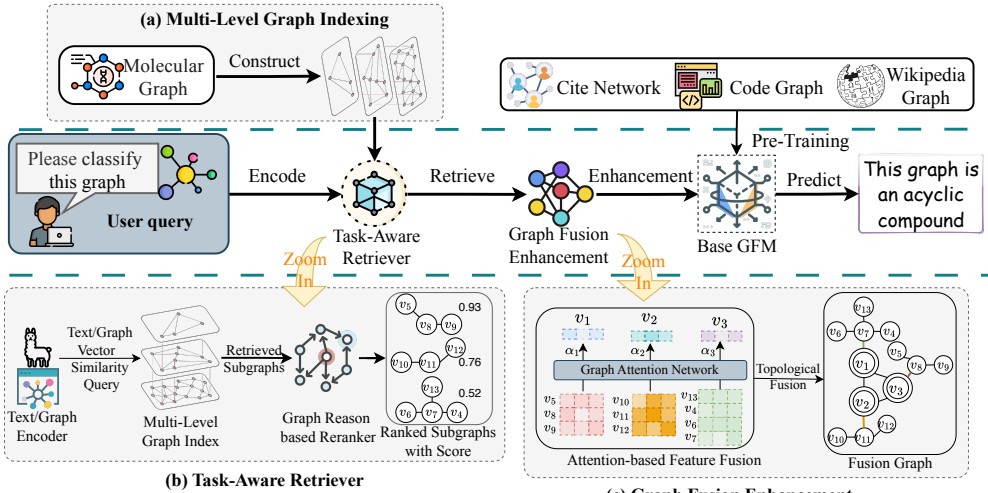

Figure 2: The overall architecture of RAG4GFM . RAG4GFM dynamically augments a Base GFM by: (a) constructing a "Multi-Level Graph Index" (b) retrieving and reranking task-relevant subgraphs based on the user query via a "Task-Aware Retriever"; and (c) fusing the retrieved subgraphs with query information through the "Graph Fusion Enhancement module", before generation.

## 3.1 Multi-level Graph Indexing

The effective application of the RAG paradigm in GFMs hinges on constructing graph data indexing systems that preserve graph completeness while ensuring efficient querying. Unlike traditional LLMs, user queries in GFM application scenarios typically present a hybrid form of text and graph data. The complex topology and multimodal attributes of graph data pose a significant challenge: designing an index system that simultaneously maintains graph structural fidelity and supports efficient retrieval. To address this challenge, we propose a multi-level graph indexing approach that first encodes mixed graph features, and then constructs hierarchical indices to enable scalable querying and retrieval.

**Mixed Feature Encoding Mechanism.** To comprehensively represent the multidimensional characteristics of graph data, this work designs four complementary feature encoding mechanisms: node feature encoding, structural feature encoding, edge feature encoding, and graph-level encoding, which align with the hierarchical reasoning capabilities discussed in Section 1. This mixed encoding strategy realizes multi-granular representations of graph data, thereby accommodating diverse retrieval requirements under varied tasks.

*Node Feature Encoding*: For nodes in the graph, we employ pre-trained language models to semantically encode the textual descriptions of node $v$ as $\mathbf{h}_t(v) = \mathrm{LM}(v)$. This method fully leverages the advantages of pre-trained language models in semantic understanding, mapping the textual information of nodes to high-dimensional semantic spaces.

*Structural Feature Encoding*: The topological position of nodes in a graph contains rich structural information. We combine Laplacian Positional Encoding (LAPPE) with node degree metrics to construct structure-aware feature representations as

$$\mathbf{h}_s(v) = \mathrm{LAPPE}(v) \oplus \mathrm{Deg}_{\mathrm{IN}}(v) \oplus \mathrm{Deg}_{\mathrm{OUT}}(v), \qquad (1)$$

where $\mathrm{LAPPE}(v)$ is the positional encoding based on the eigendecomposition of the graph laplacian matrix, $\mathrm{Deg}_{\mathrm{IN}}(v)$ and $\mathrm{Deg}_{\mathrm{OUT}}(v)$ represent the in-degree and out-degree of node $v$, respectively, and $\oplus$ denotes feature concatenation. Laplacian positional encoding effectively captures the position information of nodes in the global graph structure, while node degree information reflects local connection patterns. This encoding method is invariant to node permutations and graph isomorphism, accurately capturing nodes' relative positions within the topology.

*Edge Feature Encoding*: To capture the structural properties and topological roles of edges, we transform the original graph into its corresponding line graph. We then compute the LapPE for the nodes in this line graph. This approach allows us to generate a feature representation $\mathbf{h}_e(u, v)$ for each edge $e$ in the original graph that effectively encodes its structural context within the overall

graph topology. This approach effectively differentiates edges by their connectivity patterns and structural roles within the graph.

*Graph-level Encoding*: To obtain a holistic representation, we integrate node features, edge features, and graph statistical features:

$$\mathbf{h}_g(G) = \mathbf{h}_{\text{NODE}}(G) \oplus \mathbf{h}_{\text{EDGE}}(G) \oplus \mathbf{h}_{\text{STATS}}(G), \tag{2}$$

$$\text{where} \quad \mathbf{h}_{\text{NODE}}(G) = \text{MEAN}\{\mathbf{h}_t(v) \mid v \in V\}, \quad \mathbf{h}_{\text{EDGE}}(G) = \text{MEAN}\{\mathbf{h}_e(u,v) \mid (u,v) \in E\},$$

$$\text{and} \quad \mathbf{h}_{\text{STATS}}(G) = [|V|, |E|, \rho(G)],$$

where $\mathbf{h}_{\text{NODE}}(G)$, $\mathbf{h}_{\text{EDGE}}(G)$, and $\mathbf{h}_{\text{STATS}}(G)$ represent node feature aggregation, edge feature aggregation, and graph statistical features, respectively. $\rho(G)$ denotes the graph density, i.e., the ratio between actual and maximal possible edge counts. This comprehensive representation method fully captures the overall characteristics of the graph, providing effective support for graph-level tasks.

**Hierarchical Index Construction.** With the mixed features obtained, we proceed to organize them into an efficient multi-level index structure. This research selects the Hierarchical Navigable Small World (HNSW) as the theoretical foundation for the index structure. The hierarchical search pattern of HNSW aligns well with the multi-scale structure inherent in graph data. Accordingly, we construct a four-level hierarchical index structure covering node-, structure-, edge-, and graph-level representations: $\boldsymbol{H}_{\text{INDEX}} = \{\mathbf{h}_t(v), \mathbf{h}_s(v), \mathbf{h}_e(u,v), \mathbf{h}_g(G)\}$.

This multi-space index design offers two primary advantages: firstly, it overcomes the limitations of single representation methods by comprehensively capturing both semantic and structural graph information while flexibly supporting diverse node, edge, and graph-level retrieval tasks through a unified interface; secondly, it guarantees efficient $O(\log_2 N)$ retrieval time, crucial for large-scale graph data.

### 3.2 Task-aware Retrieval

GFMs span diverse application scenariossuch as node classification, link prediction, and graph classificationeach with distinct retrieval requirements. Traditional unified retrieval strategies fail to capture the heterogeneity of such tasks, often resulting in suboptimal efficiency and precision. To address this, we propose a task-aware retrieval framework that dynamically adapts retrieval strategies according to task characteristics, thereby improving both retrieval accuracy and computational efficiency.

**Retrieval Strategy Selector.** To accommodate heterogeneous graph tasks, the retrieval module must interpret user intent and select optimal retrieval strategies accordingly. We employ an LM-based task classifier that performs joint analysis of the natural language query and its associated graph context to predict the task type: $\tau(q) \in \{\text{NODE}, \text{EDGE}, \text{GRAPH}\}$, where $\tau$ denotes the LLM-implemented task discrimination function, and $q$ represents the user's natural language query. The predicted task type determines which feature spaces and retrieval operators are subsequently activated.

**Hybrid Feature Retrieval.**

Based on task classification results, the retrieval module adaptively selects task-relevant feature spaces to minimize irrelevant noise and align with downstream GFM objectives.

For node-level tasks, such as node classification and node regression, we jointly query both node and structural feature spaces to capture fine-grained local semantics. For edge-level tasks, including link prediction and edge classification, we leverage node and edge features to represent pairwise relational semantics. For graph-level tasks, such as graph classification and regression, we utilize holistic graph embeddings augmented with aggregated node features.

To obtain the final retrieval score for a query $q$ under the graph context $\mathcal{C}$, we apply a reciprocal rank fusion (RRF) strategy to aggregate results from multiple retrieval channels:

$$S(q|\mathbf{h}_q, \mathcal{C}) = \sum_{q \in \mathcal{F}(\mathbf{h}_q, \mathcal{C})} \frac{1}{d + \text{rank}_q^k(\mathcal{C})}, \tag{3}$$

where $\mathcal{F}(\mathbf{h}_q, \mathcal{C})$ denotes the retrieved feature set, $d$ is a smoothing constant, and $\text{rank}_x^k(\mathcal{C})$ represents the rank of candidate $x$ among the top-$k$ results. The corresponding feature sets for different tasks

are defined as:

$$\mathcal{F}(\mathbf{h}_q, \mathcal{C}) = \begin{cases} \{\mathbf{h}_t(v), \mathbf{h}_s(v)\}, & \text{if } \mathcal{C} = v \text{ (node-level)}; \\ \{\mathbf{h}_t(u), \mathbf{h}_t(v), \mathbf{h}_e(u,v)\}, & \text{if } \mathcal{C} = (u,v) \text{ (edge-level)}; \\ \{\mathbf{h}_t(v), \mathbf{h}_g(G)\}, & \text{if } \mathcal{C} = G \text{ (graph-level)}. \end{cases} \quad (4)$$

Unlike conventional methods that rely solely on node-level feature aggregation, our hybrid retrieval framework integrates semantic and structural cues across multiple levels, enabling task-adaptive retrieval that improves both precision and efficiencykey challenges emphasized in the introduction.

Unlike traditional methods that rely solely on node feature aggregation, our approach constructs more comprehensive graph representations through the integration of multi-level information.

### 3.3 Graph Fusion Enhancement

In conventional RAG frameworks for NLP tasks, retrieved text fragments are typically fused via simple concatenation or mean pooling of embeddings. However, when applied to graph data, such approaches fail to preserve topological relationships and path dependencies, resulting in substantial loss of structural information. Moreover, node importance in graphs depends heavily on positional and connectivity patterns, which traditional fusion methods cannot effectively capture. To address these limitationsparticularly the structure-semantics imbalance highlighted in Section 1, we propose a dual graph fusion enhancement mechanism, consisting of (1) attention-based feature fusion and (2) topological structure enhancement.

**Attention Feature Fusion.** We introduce a similarity-based dynamic weighting mechanism that adaptively determines the importance of each retrieved graph according to its semantic similarity with the query graph. This mechanism enables the model to assign higher weights $\alpha_i^F$ to more relevant retrieved graphs, thereby focusing on knowledge most pertinent to the query.

$$\mathbf{h}'^N_v = \mathbf{h}_v + \sum_{i=1}^{k} \left( \alpha_i^F \cdot \frac{1}{|V_i|} \sum_{u \in V_i} \mathbf{h}_u \right) \cdot \mathbb{I}[\alpha_i^F > \gamma], \qquad \text{(node-level)} \qquad (5)$$

$$\mathbf{h}'^E_{\text{SRC}} = \mathbf{h}_{\text{SRC}} + \sum_{i=1}^{k} \left( \alpha_i^F \cdot \frac{1}{|V_i|} \sum_{u \in V_i} \mathbf{h}_u \right) \cdot \mathbb{I}[\alpha_i^F > \gamma], \qquad \text{(edge-level, source)} \qquad (6)$$

$$\mathbf{h}'^E_{\text{DST}} = \mathbf{h}_{\text{DST}} + \sum_{i=1}^{k} \left( \alpha_i^F \cdot \frac{1}{|V_i|} \sum_{u \in V_i} \mathbf{h}_u \right) \cdot \mathbb{I}[\alpha_i^F > \gamma], \qquad \text{(edge-level, destination)} \qquad (7)$$

$$\mathbf{h}'^G_v = \frac{1}{k} \sum_{i=1}^{k} \left( \alpha_i^F \cdot \beta_v^i \cdot \mathbf{h}_v \right) \cdot \mathbb{I}[\alpha_i^F > \gamma], \qquad \text{(graph-level)} \qquad (8)$$

$$\beta_v^i = \text{SOFTMAX}(\mathbf{h}_v \cdot (\mathbf{h}_g^q + \mathbf{h}_g^i)), \qquad \text{(graph-level, attention)} \qquad (9)$$

where $\mathbf{h}'^N_v$ is the enhanced target node feature, $\mathbf{h}_v$ is the original node feature, $\mathbf{h}'^E_{\text{SRC}}$ and $\mathbf{h}'^E_{\text{DST}}$ are the enhanced source and destination edge features, $\mathbf{h}'^G_v$ is the enhanced target graph feature, $V_i$ is the node set of the $i$-th retrieved graph, and $\gamma$ is the feature fusion threshold (default value 0.5) used to filter low-relevance retrieved results. $\mathbf{h}_g^q$ and $\mathbf{h}_g^i$ represent the global features of the query graph and the $i$-th retrieved graph, respectively, and $k$ is the number of effective retrieved results.

Compared to traditional feature fusion methods, the attention feature fusion mechanism proposed in this research adaptively emphasizes highly relevant knowledge through dynamic weight allocation, effectively reducing noise impact while utilizing threshold filtering mechanisms to avoid interference from low-quality retrieved results.

**Topological Structure Enhancement.** Complementary to feature fusion, the topological structure enhancement module focuses on enriching the query graph's connectivity patterns. It selectively incorporates structurally relevant edges from retrieved graphs, weighted by their graph-level relevance scores $\alpha_i^T$, thereby augmenting the query graph with semantically aligned structural information. Implemented via sparse adjacency operations, this module efficiently handles large-scale graph data while maintaining topological consistency.

Given the adjacency matrix $\boldsymbol{A}$ of the query graph and the set of adjacency matrices $\boldsymbol{A}_1, \boldsymbol{A}_2, ..., \boldsymbol{A}_k$ of retrieved graphs, the enhanced adjacency matrix is computed as:

$$\boldsymbol{A}' = \boldsymbol{A} + \sum_{i=1}^{k} \alpha_i^T \cdot \boldsymbol{A}_i \cdot \mathbb{I}[\alpha_i^T > \gamma], \tag{10}$$

where $\alpha_i^T$ serves as the relevance weight. To suppress noisy edges, we apply threshold filtering such that $\boldsymbol{A}'_{uv} = \mathbb{I}[\boldsymbol{A}'_{uv} > \delta]$, where $\delta$ is the edge addition threshold (default value 0.5). This sparse matrix-based implementation achieves efficient large-scale processing while preserving graph sparsity and interpretability.

**Other analyses.** Due to space limitations, we present analyses on the complexity in Appendix A.

## 4 Experiments

### 4.1 Experimental Setup

**Datasets.** To evaluate the effectiveness and generality of RAG4GFM , we conduct experiments on a diverse collection of graph datasets covering six task types. For node classification and link prediction, we adopt the latest TAG benchmark [48] to prevent potential data leakage from GFM pre-training. For other tasks, we select datasets that, to the best of our knowledge, were not used in any GFM pre-training phase, verified through public model documentation and release notes. Table 1 summarizes the dataset statistics, with additional details in Appendix B.1.

Table 1: Statistics of the dataset used in our experiments. "Task" denotes the downstream task type: NC (Node Classification), NR (Node Regression), LC (Link Classification), LP (Link Prediction), GC (Graph Classification), and GR (Graph Regression). "Graphs" indicates the number of graph instances; a value of 1 denotes a single large graph.

| Dataset | Nodes | Edges | Graphs | Domain | Task |
|---|---|---|---|---|---|
| Books-Children [48] | 76,875 | 1,554,578 | 1 | E-commerce | NC,LP |
| Books-History [48] | 41,551 | 358,574 | 1 | E-commerce | NC, LP |
| Ele-Computers [48] | 87,229 | 721,081 | 1 | E-commerce | NC, LP |
| Ele-Photo [48] | 48,362 | 500,928 | 1 | E-commerce | NC, LP |
| MiniGCDataset [49] | 21,909 | 177,875 | 1,000 | Synthetic | GC |
| BA2MotifDataset [50] | 25,000 | 51,392 | 1,000 | Synthetic | GC |
| Chameleon [51] | 2,277 | 36,101 | 1 | Wikipedia | NR |
| WN18Dataset [52] | 40,943 | 151,442 | 1 | Knowledge | LC |
| QM7bDataset [53] | 108,165 | 1,766,695 | 7,211 | Biology | GR |

**Baselines.** We evaluate RAG4GFM against state-of-the-art approaches from three categories: (1) **Prompt Engineering Methods:** Few-shot Learning [54], Chain-of-Thought (COT) [55], IR-augmented COT [56]. These methods focus on structuring the input prompt to guide the GFM's inference without altering model parameters, leveraging strategies like few-shot examples or explicit reasoning steps. (2) **Retrieval-Enhanced Methods:** VanillaRAG [30], GraphRAG [43], G-Retriever [57]. These methods augment GFMs with external knowledge retrieval and ground predictions in relevant evidence, ranging from text to graph-aware retrieval. (3) **Graph Out-of-Distribution Generalization Methods:** Prototype [58],GNNSafe [59]. These approaches focus on the ability to generalize to unseen domains, structures, or distributions, emphasizing robustness and transferability.

**GFMs.** We evaluate RAG4GFM on seven representative GFMs, grouped by their predictive architecture into three types: (1) GNNs as predictor: OpenGraph [35] and AnyGraph [17]. (2) Co-learning GNNs and LLMs: GLEM [60]. (3) LLMs as predictor: GraphGPT [16], HiGPT [61], LLaGA [62], and GraphAdapter [63]. Further details and descriptions of these models are provided in Appendix B.2.

### 4.2 Effectiveness of RAG-Enhanced GFMs

Our first research question investigates whether RAG-based mechanisms enhance the performance of existing GFMs across various graph tasks. Table 2 shows the results for node classification and

link prediction, while additional findings for node regression, link classification, graph classification, and graph regression are reported in Appendix C.1. Note that some GFMs lack results for tasks not supported by their architectures; for instance, OpenGraph does not support graph-level classification.

Table 2: Results of Node Classification and Link Prediction on Four Datasets

| Model | Datasets | | | | | | | | | | | |
| | Computers | | | History | | | Fitness | | | Photo | | |
| | Acc | ROC-AUC | Recall | Acc | ROC-AUC | Recall | Acc | ROC-AUC | Recall | Acc | ROC-AUC | Recall |
|---|---|---|---|---|---|---|---|---|---|---|---|---|
| **Node Classification Results** | | | | | | | | | | | | |
| HiGPT | 76.82 | 61.45 | 58.76 | 60.45 | 59.78 | 54.34 | 70.12 | 58.67 | 57.23 | 63.88 | 57.45 | 53.21 |
| HiGPT+RAG4GFM | 79.34 | 66.89 | 66.42 | 63.21 | 63.45 | 59.76 | 72.89 | 63.78 | 64.32 | 67.50 | 63.24 | 60.45 |
| GraphGPT | 75.45 | 62.34 | 58.23 | 59.88 | 58.90 | 53.67 | 69.34 | 59.23 | 56.89 | 62.15 | 56.78 | 52.90 |
| GraphGPT+RAG4GFM | 81.23 | 67.56 | 67.45 | 64.92 | 64.32 | 60.54 | 74.88 | 64.56 | 65.23 | 69.45 | 63.89 | 61.23 |
| GLEM | 79.45 | 63.56 | 60.12 | 54.67 | 57.23 | 52.45 | 73.89 | 60.45 | 58.90 | 61.23 | 56.23 | 53.45 |
| GLEM+RAG4GFM | 83.12 | 68.90 | 68.23 | 59.89 | 61.78 | 57.34 | 75.45 | 65.12 | 66.78 | 66.90 | 62.56 | 59.34 |
| LLaGA | 71.23 | 59.45 | 55.67 | 58.90 | 58.23 | 53.21 | 64.56 | 55.34 | 54.45 | 59.78 | 55.45 | 51.78 |
| LLaGA+RAG4GFM | 76.89 | 64.23 | 63.45 | 61.23 | 62.34 | 58.56 | 71.90 | 61.23 | 61.45 | 65.45 | 61.23 | 58.34 |
| GraphAdapter | 76.78 | 62.78 | 59.34 | 52.34 | 56.45 | 51.23 | 69.90 | 59.78 | 57.45 | 65.78 | 58.45 | 54.67 |
| GraphAdapter+RAG4GFM | 81.23 | 67.45 | 67.23 | 57.89 | 61.23 | 56.78 | 73.45 | 64.34 | 64.56 | 69.90 | 64.23 | 61.56 |
| OpenGraph | 74.67 | 61.32 | 57.73 | 62.71 | 61.58 | 55.82 | 67.59 | 57.80 | 56.74 | 60.93 | 56.97 | 52.48 |
| OpenGraph+RAG4GFM | 79.38 | 66.43 | 65.01 | 63.92 | 64.80 | 61.55 | 73.11 | 62.77 | 63.38 | 67.55 | 62.70 | 59.10 |
| AnyGraph | 73.45 | 60.89 | 57.23 | 61.89 | 60.23 | 54.90 | 67.90 | 57.45 | 55.89 | 61.23 | 56.45 | 52.67 |
| AnyGraph+RAG4GFM | 79.56 | 66.12 | 65.34 | 63.45 | 64.12 | 60.23 | 72.34 | 62.89 | 63.12 | 67.89 | 62.78 | 59.45 |
| **Link Prediction Results** | | | | | | | | | | | | |
| OpenGraph | 53.33 | 39.84 | 36.58 | 40.87 | 40.70 | 34.28 | 46.61 | 35.62 | 35.16 | 39.33 | 35.79 | 30.97 |
| OpenGraph+RAG4GFM | 58.21 | 45.18 | 43.66 | 42.27 | 44.14 | 39.92 | 52.00 | 40.71 | 42.25 | 45.82 | 41.77 | 37.76 |
| HiGPT | 55.25 | 40.68 | 37.43 | 38.99 | 38.56 | 33.13 | 49.44 | 36.61 | 35.90 | 42.06 | 36.26 | 32.05 |
| HiGPT+RAG4GFM | 58.00 | 45.85 | 45.28 | 41.94 | 43.02 | 37.76 | 52.29 | 41.78 | 43.18 | 45.93 | 42.52 | 39.17 |
| GraphGPT | 54.27 | 41.03 | 37.16 | 37.71 | 38.04 | 32.09 | 48.51 | 36.72 | 35.06 | 40.88 | 35.59 | 31.13 |
| GraphGPT+RAG4GFM | 60.37 | 46.05 | 45.93 | 42.96 | 43.62 | 39.04 | 53.14 | 42.68 | 44.27 | 47.74 | 43.06 | 39.81 |
| AnyGraph | 52.17 | 38.53 | 36.17 | 40.01 | 39.68 | 33.07 | 46.46 | 35.47 | 34.30 | 39.52 | 35.29 | 31.06 |
| AnyGraph+RAG4GFM | 58.26 | 44.92 | 43.94 | 41.74 | 43.26 | 38.84 | 51.34 | 40.59 | 41.83 | 45.97 | 41.53 | 38.12 |

Across all four datasets, the RAG4GFM yields consistent and substantial improvements, demonstrating its strong generalization capability. For instance, GLEM+RAG4GFM achieves 83.12% accuracy on node classification in the Computers dataset, representing a relative gain of approximately 5.5% over GLEM. Similarly, GraphGPT+RAG4GFM attains 60.37% accuracy on link prediction, outperforming its vanilla counterpart by 5.1%.

The improvements are consistently observed across all four datasetsComputers, History, Fitness, and Photohighlighting the broad applicability of our approach. For example, GLEM+RAG4GFM achieves 83.12% accuracy on node classification in the Computers dataset, a relative gain of 5.5% over the baseline GLEM. Similarly, GraphGPT+RAG4GFM attains 60.37% accuracy on link prediction, outperforming its vanilla counterpart by 5.1%. These consistent trends indicate that RAG-based augmentation benefits both GNN-based and LLM-based GFMs. To elucidate the source of these improvements, we analyze the core design of RAG4GFM . The *Multi-level Indexing*, coupled with the *Task-Aware Retriever* module, ensures that each GFM retrieves the most relevant external subgraphs and feature spaces efficiently, reducing noise and retrieval latency. Finally, the *Graph Fusion Enhancement* module integrates retrieved information into the query graph through both feature-level and topology-level fusion. This integration refines node embeddings for classification, strengthens edge inference for link prediction, and provides more coherent context for graph-level reasoning. These components operate synergisticallyencoding richer local context, retrieving task-relevant external knowledge, and integrating it in a structure-consistent manner.

Overall, the results demonstrate that RAG4GFM effectively overcomes the static-parameter limitation of conventional GFMs by dynamically grounding predictions in external structured knowledge. This design directly addresses the integration bottleneck highlighted in the introduction, leading to more accurate, contextually grounded, and robust graph-level reasoning across tasks without requiring additional post-training.

## 4.3 Ablation Study on RAG4GFM

To quantitatively evaluate the individual contributions of RAG4GFM 's key components, we perform a series of ablation experiments using **AnyGraph** as the base GFM under identical hyperparameter settings for fair comparison. We analyze both architectural modules (retrieval, fusion, indexing)

and feature-level encodings to understand how each design choice affects performance on node classification (Computers) and link prediction (History) tasks.

**Variants on Core Architecture. w/o RAG.** Removes the entire retrieval-augmented generation (RAG) pipeline; the GFM operates without external knowledge retrieval or integration. **w/o GF.** Retains retrieval but replaces the specialized graph fusion module with a naive concatenation strategy, removing the semantic-structural alignment mechanism. **w/o GI.** Replaces the hierarchical graph indexing with a basic text-similarity index, assessing the importance of structure-aware retrieval.

**Variants on Feature Encoding.** To further assess the effectiveness of our structural feature design, we compare three encoding strategies: **LAPPE only.** Uses Laplacian positional encodings, capturing global structural information but limited in local awareness. **Node Degree only.** Uses node in/out-degree as simple local centrality features. **LAPPE + Degree.** Combines both global positional and local topological cues, aligning with the principle of global-local structural complementarity.

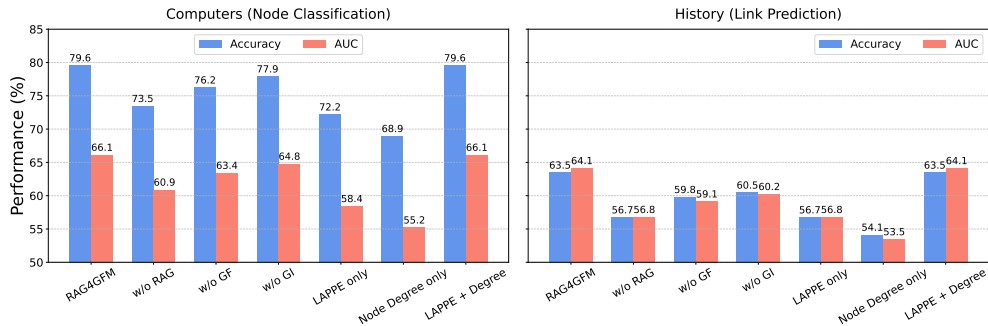

Figure 3: Ablation study results for RAG4GFM on node classification (Computers) and link prediction (History) tasks. Higher values are better.

As illustrated in Figure 3, the full RAG4GFM configuration consistently achieves the best results across both tasks. Among the architectural variants, *w/o RAG* exhibits the largest drop, confirming that retrieval is crucial for integrating dynamic external knowledge. Removing the graph fusion module (*w/o GF*) also decreases accuracy and AUC, showing that our fusion mechanism is essential for coherent reasoning beyond naive feature concatenation. The performance reduction of *w/o GI* highlights the role of hierarchical, structure-aware indexing in precise, low-noise retrieval.

Regarding feature encoding, models using only LAPPE or only node degree perform noticeably worse than the joint configuration. The combined LAPPE+Degree variant recovers nearly the same performance as the full model, validating that global-local structural complementarity enhances graph representation learning. This observation is consistent across both Computers and History tasks, underscoring that balanced structural cues benefit retrieval and fusion.

All componentsRAG retrieval, graph fusion, hierarchical indexing, and global-local feature encodingcontribute jointly to RAG4GFM 's effectiveness. Their synergy enables efficient updates, faithful reasoning, and more accurate predictions across diverse graph tasks.

## 4.4 Comparison with Other Knowledge Updating Methods

We evaluate tasks where external knowledge is critical and frequently updated, such as knowledge-intensive node classification.

Results in Table 3 show that RAG4GFM consistently outperforms all competing approaches, confirming its capability to retrieve and integrate relevant external information for accurate prediction. Retrieval-enhanced (RE) methods generally outperform prompt-engineering (PE) and graph out-of-distribution (G-OOD) methods. Among RE baselines, graph-aware models such as GraphRAG and G-Retriever achieve stronger results than lexical retrievers, yet RAG4GFM delivers further gains by explicitly fusing semantic and structural representations. PE methods provide only marginal improvements, while text-only augmentation (e.g., IR-augmented CoT) remains insufficient for complex graph reasoning. G-OOD-oriented approaches (Prototype, GNNSafe) underperform in our setting, as they emphasize distributional robustness rather than dynamic knowledge updating.

Table 3: Performance comparison of RAG4GFM with knowledge updating baselines on node classification(Computers), link prediction (History), and graph classification (MiniGCDataset).

| Category | Method | Computers (Node Class.) | | History (Link Pred.) | | MiniGCDataset (Graph Class.) | |
|---|---|---|---|---|---|---|---|
| | | Accuracy | ROC-AUC | Accuracy | ROC-AUC | Accuracy | ROC-AUC |
| Prompt Engineering | Few-shot Learning [54] | 69.62 | 59.88 | 58.27 | 56.35 | 59.24 | 46.26 |
| | COT [55] | 69.73 | 62.46 | 59.28 | 56.71 | 59.56 | 49.47 |
| | IR-augmented COT [56] | 69.68 | 63.62 | 58.58 | 59.46 | 59.41 | 48.59 |
| Graph Out-of-Distribution | Prototype [58] | 71.04 | 60.77 | 60.15 | 56.86 | 60.03 | 47.48 |
| | GNNSafe [59] | 70.89 | 63.56 | 59.62 | 61.44 | 60.57 | 49.42 |
| Retrieval-Enhanced | VanillaRAG [30] | 71.58 | 64.63 | 61.42 | 60.57 | 63.46 | 51.58 |
| | GraphRAG [43] | 74.47 | 64.61 | 59.84 | 61.55 | 63.18 | 51.86 |
| | G-Retriever [57] | 75.06 | 64.95 | 60.27 | 61.97 | 63.73 | 52.32 |
| **RAG4GFM (Ours)** | AnyGraph-based | **79.56** | **66.12** | **63.45** | **64.12** | **65.51** | **51.88** |

In summary, RAG4GFM demonstrates clear advantages as a knowledge-enhancement framework. Compared with traditional fine-tuning or static knowledge-graph integration, the RAG paradigm offers a flexible and efficient mechanism for updating GFMs with external knowledge. It achieves this without costly retraining while maintaining reasoning fidelity.

Beyond predictive performance, we further examine the efficiency of RAG4GFM compared to GraphLoRA [22], a representative parameter-efficient fine-tuning (PEFT) approach. Table 4 reports the time and GPU memory required to reach identical accuracy targets. RAG4GFM achieves comparable accuracy while reducing runtime by up to **7.0×** and GPU memory usage by over **60%**. This efficiency arises from its retrieval-based updating paradigm, which avoids gradient-based optimization and large parameter storage. Moreover, since RAG4GFM refreshes knowledge through lightweight retrieval and fusion rather than re-training, it scales favorably to frequent graph updates and resource-constrained environments.

Table 4: Efficiency and memory comparison between RAG4GFM and GraphLoRA [22] under identical accuracy targets. Reported metrics include training time and peak GPU memory usage.

| GFM | Dataset | Target Acc | Time | Peak GPU Memory |
|---|---|---|---|---|
| GraphLoRA + AnyGraph | Computers | 78% | 7.32 Hours | 25.23 GB |
| RAG4GFM + AnyGraph | Computers | 78% | 63 Minutes | 9.86 GB |
| GraphLoRA + HiGPT | History | 63% | 5.87 Hours | 17.18 GB |
| RAG4GFM + HiGPT | History | 63% | 19 Minutes | 5.15 GB |

**Other analyses.** Due to space limitations, we present complexity analysis, experimental setups, and further experimental results in C.

## 5 Conclusions

In this paper, we introduce RAG4GFM , a RAG framework tackling two critical GFM challenges: efficient knowledge updating and faithful reasoning. Leveraging a three-component architecture, RAG4GFM achieves significant gains in knowledge-update efficiency and reasoning faithfulness over traditional parameter-updating approaches, as demonstrated by extensive empirical evaluation across diverse domains. In future work, we will focus on: scalability to billion-node graphs and real-time systems via disk-based ANN (e.g., DiskANN) and asynchronous fusion; and broader directions, including leveraging negative/contrastive knowledge to refine decision boundaries and extending the framework to multimodal graphs that include images. Our goal is to promote responsible AI practices in the development and deployment of RAG-enhanced GFMs.

## Acknowledgments and Disclosure of Funding

This research is supported by the Open Project of Key Laboratory of Industrial Software Engineering and Application Technology, Ministry of Industry and Information Technology (HK202403641), the National Science Foundation of China under Grant 62125206, the Zhejiang Provincial Natural Science Foundation of China under Grant LQ24F020019, and the Hangzhou Key Research and Development Program under Grant 2025SZD1A03.

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

# Appendix

## A   Model analysis

### A.1   Complexity Analysis

Computational efficiency is crucial for the practical deployment of RAG4GFM in large-scale graph scenarios. This section analyzes the time complexity and space complexity of its key components.

**Multi-level Graph Indexing:** The HNSW-based hierarchical graph index is constructed with time complexity $O(N \log N)$, where $N$ is the number of graphs or nodes. The index occupies $O(N)$ space, with constants determined by parameters such as the maximum connection number $M$ (typically $M \in [16, 64]$).

**Task-aware Retriever:** For a single query, the hierarchical retrieval process operates in $O(\log N)$ timesignificantly faster than the $O(N)$ cost of linear searchwhile requiring only $O(k)$ auxiliary space for $k$ retrieved candidates (typically $k \ll N$).

**Graph Fusion Enhancement:** In the fusion stage, attention-based feature aggregation has time complexity $O(k|V|d)$, where $|V|$ is the average number of nodes per retrieved subgraph and $d$ the feature dimension. The subsequent structural enhancement costs $O(k|V|^2)$ for dense graphs or $O(k|E|)$ when sparse adjacency lists are used. The dominant space cost stems from storing the fused graph representation, typically $O(|V|^2)$ for dense adjacency.

**Overall Pipeline:** Combining these components, the per-query time complexity is $O(\log N + k|V|^2)$ for dense and $O(\log N + k|E|)$ for sparse graphs. The total space complexity is $O(N + |V|^2)$, comprising the index storage ($O(N)$) and the fused graph representation ($O(|V|^2)$).

## B   Experimental Setups

### B.1   Datasets

We evaluate on diverse datasets spanning e-commerce, synthetic, molecular, and knowledge graph domains. (1) *Books-Children*, *Books-History* and *Ele-Computers* and *Ele-Photo* [48] are Amazon co-purchase graphs of books and electronics. (2) *MiniGCDataset* [49] and *BA2MotifDataset* [50] are synthetic datasets used by Barabasi-Albert model. (3) *QM7bDataset* [53, 64] contains ∼7,000 stable organic molecules with computed quantum properties. (4) *WN18Dataset* [52] is a multi-relational dataset extracted from Wordnet. (5) *Chameleon* [51, 65] is wikipedia page-page network.

### B.2   Graph Foundation Models

We used seven representative GFMs as backbones for our experiments: (1) *OpenGraph* [35]: aims to establish an open-domain GFM exhibiting strong zero-shot generalization by distilling LLM knowledge and employing a unified graph tokenizer adaptable to unseen graph structures. (2) *AnyGraph* [17]: addresses the challenge of graph heterogeneity by proposing a Mixture-of-Experts GNN architecture with automated routing, enabling efficient generalization and adaptation across diverse graph domains. (3) *GraphGPT* [16]: focuses on aligning LLMs with graph structures through instruction tuning, thereby enabling LLMs to comprehend, reason about, and generalize across various graph-based tasks in a zero-shot manner. (4) *HiGPT* [61]: extends the LLM alignment concept specifically to heterogeneous graphs, utilizing instruction tuning techniques and a context-aware tokenizer to make LLMs proficient in understanding diverse node and relation types. (5) *GraphAdapter* [63]: introduces a parameter-efficient approach to integrate graph reasoning into frozen LLMs by inserting and fine-tuning lightweight GNN adapter modules, bridging structural and textual information without full retraining. (6) *LLaGA* [62]: enables frozen LLMs to directly ingest and process graph information by transforming graph structures into specialized node sequences via templates and mapping them through a versatile projector into the LLM's embedding space. (7) *GLEM* [60]: proposes a synergistic co-learning framework based on expectation-maximization, allowing GNNs and LMs to iteratively enhance each other's representations and predictions on text-attributed graphs

## B.3 Settings and Parameters

To evaluate the enhancement provided by RAG4GFM to GFMs, we adopted a zero-shot evaluation protocol. Under this protocol, GFMs perform inference on test datasets without task-specific fine-tuning, leveraging contextual information supplied by RAG4GFM . Specifically, during its retrieval stage, RAG4GFM identifies and provides the three subgraphs yielding the highest relevance probabilities to inform the GFM's subsequent processing for each task instance.

Performance is quantified using standard metrics: Accuracy (Acc), Recall, and ROC-AUC are employed for classification and link prediction tasks, while Mean Squared Error (MSE) and the Coefficient of Determination ($R^2$) are applied for regression tasks.

All experiments are conducted on a server equipped with an AMD EPYC 7B12 CPU, an NVIDIA A6000 GPU, and 512GB of RAM. The software environment comprised Ubuntu 22.04, Python 3.10, PyTorch 2.2, and DGL 2.3.

## C Further Experimental Results

### C.1 Effectiveness of RAG-Enhanced GFMs

The experimental results are shown in Table 5. Among them, the missing model data indicates that this model does not support this task.

Table 5: Results of Graph Classification Tasks on Two Datasets

| Model | MiniGCDataset | | | BA2MotifDataset | | |
|---|---|---|---|---|---|---|
| | Acc | ROC-AUC | Recall | Acc | ROC-AUC | Recall |
| HiGPT | 62.44 | 47.86 | 44.03 | 46.33 | 44.98 | 40.06 |
| HiGPT+RAG4GFM | 64.71 | 53.32 | 51.89 | 49.28 | 49.22 | 44.91 |
| GraphGPT | 60.98 | 48.03 | 44.45 | 44.71 | 44.25 | 39.37 |
| GraphGPT+RAG4GFM | 67.61 | 52.95 | 52.70 | 50.30 | 49.87 | 46.47 |
| GLEM | 56.58 | 45.51 | 40.69 | 44.57 | 42.63 | 37.61 |
| GLEM+RAG4GFM | 61.76 | 50.27 | 48.83 | 47.41 | 48.55 | 43.49 |
| AnyGraph | 58.67 | 46.25 | 42.78 | 47.53 | 45.88 | 40.50 |
| AnyGraph+RAG4GFM | 65.51 | 51.88 | 51.27 | 48.54 | 50.23 | 46.31 |

Table 6: Supplementary Results for Node Regression, Link Classification, and Graph Regression Tasks

| Model | Chameleon (Node Reg.) | | WN18Dataset (Link Class.) | | | QM7bDataset (Graph Reg.) | |
|---|---|---|---|---|---|---|---|
| | MSE | $R^2$ | Acc | Recall | ROC-AUC | MSE | $R^2$ |
| AnyGraph | 6.83 | 0.55 | 68.50 | 63.27 | 72.94 | 10.52 | 0.65 |
| AnyGraph+RAG4GFM | 1.47 | 0.63 | 73.05 | 68.75 | 77.40 | 3.42 | 0.73 |

Table 5 reports graph-classification results on MiniGCDataset and BA2MotifDataset, and Table 6 summarizes supplementary results on node regression (Chameleon), link prediction (WN18), and graph regression (QM7b). Across all backbones listed in Table 5, integrating RAG4GFM consistently improves Accuracy, ROC-AUC, and Recall. For example, on MiniGCDataset, AnyGraph+RAG4GFM increases Accuracy from 58.67% to 65.51%, and GraphGPT+RAG4GFM from 60.98% to 67.61%. Similar gains appear on BA2MotifDataset (e.g., GraphGPT+RAG4GFM raises ROC-AUC from 44.25% to 49.87%). These results indicate that the retrieved subgraphs provide complementary structural/semantic cues that strengthen graph-level representations.

For the additional tasks in Table 6, RAG4GFM reduces MSE and increases $R^2$ on Chameleon (node regression) and QM7b (graph regression), and improves all metrics on WN18 (link prediction). These trends are consistent with our findings on classification and support the conclusion that retrieval-based augmentation enhances both local (node-level) and global (graph-level) predictive signals.

Overall, the additional tasks corroborate the broad applicability of RAG4GFM and its ability to dynamically enrich GFMs with external graph-structured knowledgedirectly addressing the static-knowledge bottleneck highlighted in the introduction.

## C.2  Hyperparameter Robustness Analysis

We evaluate the robustness and deployability of RAG4GFM by examining three key hyperparametersretrieval size ($K$), feature fusion threshold ($\gamma$), and edge addition threshold ($\delta$).

**Effect of Retrieval Size $K$.** We study the impact of $K$ in $\{1, 2, 3, 4, 5\}$ and summarize the results in Figure 4. Performance improves steadily as $K$ increases from 1 to 3, since additional retrieved subgraphs enrich contextual reasoning. Beyond this point, accuracy plateaus or slightly declines because excessive retrieval introduces redundancy and noise. Balancing accuracy and efficiency, we set $K = 3$ as the default in all experiments.

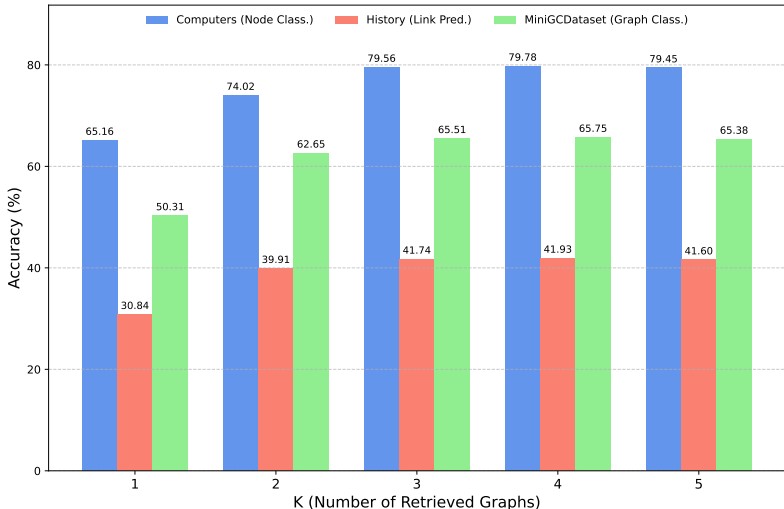

Figure 4: Performance of RAG4GFM with different numbers of retrieved subgraphs ($K$).

**Sensitivity to Feature and Structure Fusion Thresholds.**

We further investigatethe sensitivity of RAG4GFM to the feature fusion relevance threshold $\gamma$ and the structural pruning threshold $\delta$. Both are varied within $\{0.3, 0.5, 0.7, 0.9\}$ using Any-Graph+RAG4GFM as a representative backbone on node classification task.

Table 7: Sensitivity of RAG4GFM to feature ($\gamma$) and structure ($\delta$) fusion thresholds. Node classification accuracy (%).

| $\gamma\backslash\delta$ | 0.3 | 0.5 | 0.7 | 0.9 |
|---|---|---|---|---|
| 0.3 | 76.23 | 77.89 | 78.12 | 75.67 |
| 0.5 | 78.45 | **79.56** | 79.28 | 76.91 |
| 0.7 | 78.73 | 79.14 | 78.85 | 76.34 |
| 0.9 | 75.12 | 76.47 | 75.98 | 73.89 |

AnyGraph achieves its best accuracy when $\gamma$ and $\delta$ are around $0.5$, indicating a balanced trade-off between incorporating sufficient external evidence and filtering noise. Very low thresholds ($0.3$) allow irrelevant retrievals, while overly strict ones ($0.9$) remove useful structural links, leading to slight degradation. These results show that RAG4GFM is robust to small parameter variations and requires minimal tuning.

AnyGraph achieves its best accuracy when $\gamma$ and $\delta$ are around $0.5$, indicating a balanced trade-off between incorporating sufficient external evidence and filtering noise. Very low thresholds ($0.3$) allow irrelevant retrievals, while overly strict ones ($0.9$) remove useful structural links, leading to slight degradation. These results show that RAG4GFM is robust to small parameter variations and requires minimal tuning.

Across all hyperparameters, RAG4GFM demonstrates stable behavior and clear optimal regions. Moderate settings (e.g., $K = 3$, $\gamma = \delta = 0.5$) achieve strong accuracy without costly parameter

search, validating the framework's robustness and practicality for dynamic graph environmentsdirectly addressing the scalability and maintainability challenges highlighted in the introduction.

### C.3 Case Study: Link Prediction in Product Co-Purchase Networks

To concretely illustrate the efficacy of RAG4GFM 's retrieval and fusion mechanisms for link prediction, we present a case study on the Amazon Computers dataset. In this dataset, nodes represent products, and an edge between two products indicates they are frequently bought together. Product features are derived from bag-of-words representations of their reviews. The task is to predict whether two products, not currently linked, are likely to be co-purchased.

**Scenario.** Consider two products:

- Product A: "SpectrePro X15 Ultrabook"reviews emphasize portability, sleek design, and long battery life, suggesting usage by mobile professionals.
- Product B: "PowerStation Multi-Port Hub"reviews highlight multiple ports and suitability for complex desktop setups, often used for stationary, high-performance workstations.

A base GFM relying solely on these textual features predicts a low probability of co-purchase, since the two items appear to target distinct user needs.

**Contextual Graph Retrieval.** RAG4GFM performs retrieval over the existing Amazon Computers co-purchase graph to find contextual subgraphs for the pair (A, B). For Product A, retrieved subgraphs include portable accessories (e.g., "wireless mouse", "laptop sleeve") and similar thin-and-light ultrabooks. For Product B, they include desktop monitors, ergonomic keyboards, and other workstation components. Crucially, RAG4GFM also retrieves bridging subgraphs connecting ultrabooks that are frequently co-purchased with docking stations or multi-port hubs. These subgraphs reveal usage patterns where high-end ultrabooks often serve dual rolesas portable devices and as central elements of larger workstation setups. Such retrieved evidence enriches the context beyond the isolated features of A and B.

**Information Fusion for Enhanced Representation.** The fusion module integrates the original node features of the "SpectrePro X15 Ultrabook" and the "PowerStation Hub" with the structural and semantic cues from the retrieved subgraphs. This allows the GFM to recognize that, despite distinct marketing positions, both products cater to overlapping user behaviors requiring mobility and connectivity. The fused representation refines node embeddings and relational cues, bridging the gap between portable and stationary usage scenarios.

**Model Inference.** With this context-enriched representation, the GFM within RAG4GFM re-evaluates the co-purchase likelihood for the product pair and assigns a significantly higher probability of a link. By leveraging retrieved graph context, the model overcomes superficial feature dissimilarity and bases its prediction on evidence of similar co-purchase behaviors found in the broader network.

**Insight.** This case study demonstrates that by dynamically retrieving and fusing relevant subgraphs, RAG4GFM enables GFMs to uncover non-obvious relationships and make more accurate link predictions. It highlights the ability to move beyond direct node attributes and leverage network-level patternsan essential capability for understanding complex co-purchase dynamics in real-world graph applications.

