# OpenReview forum: "RAG4GFM: Bridging Knowledge Gaps in Graph Foundation Models through Graph Retrieval Augmented Generation"
_NeurIPS.cc/2025/Conference — NeurIPS 2025 oral_

### Official Review · Reviewer_xzY6 · 2025-06-30

**Clarity:** 2
**Significance:** 2
**Originality:** 2
**Rating:** 4
**Confidence:** 4

**Summary:**

This paper proposes a RAG framework tailored for Graph Foundation Models (GFMs). The authors design a hierarchical graph indexing system that captures semantic and structural information at node, edge, and graph levels. A task-aware retrieval strategy is introduced to retrieve relevant subgraphs based on the nature of the query. Retrieved information is then integrated through a graph fusion enhancement mechanism to support downstream graph reasoning tasks. Extensive experiments are conducted across various GFM architectures and graph learning tasks, demonstrating the effectiveness of the proposed RAG4GFM framework compared to non-RAG-enhanced GFMs.

**Questions:**

1.  Please explicitly specify the external retrieval sources used for each task and dataset. While Figure 1 mentions Code Graph, Cite Network, and Wikipedia, it is not clear whether these were actually used in the experiments. Provide statistics on the source size, number of indexed subgraphs, and how these sources were constructed and queried.

2.  Present qualitative examples of retrieved subgraphs and how they influence predictions. This would help demonstrate the framework’s interpretability and provide evidence of effective knowledge grounding.

3. Consider evaluating the contribution of each index level (node, edge, graph) and each component of the Graph Fusion Enhancement module. This would help justify the design choices and confirm the necessity of all components.

4. Section 3.3 is difficult to follow. Provide a clearer explanation of the attention-based fusion and structural enhancement, preferably with a simplified diagram or concrete example that aligns with Equations (4)–(9).

5. The provided code is incomplete, which lacks direct support from GFM directory or instruction.

**Ethical Concerns:**

["NO or VERY MINOR ethics concerns only"]

**Final Justification:**

As I mentioned in rebuttal phase, the authors have resolved my concerns on model design, ablation study, etc, so I will raise my evaluation score

**Limitations:**

The authors acknowledge limitations in the conclusion, but the discussion is quite general and lacks depth. A more concrete discussion of technical limitations and future mitigation strategies would enhance the value of the work.

**Quality:**

3

**Strengths And Weaknesses:**

### Strengths
* The paper is clearly structured and generally easy to follow.
* The investigated research problem is interesting and important to the graph learning community.
* The proposed framework is evaluated on a broad range of tasks (node classification, link prediction, etc.) and datasets, which improves validation.

### Weaknesses
- The core architectural design largely adapts existing concepts from RAG in LLMs to GFMs. The hierarchical indexing and fusion strategies are also based on known graph learning practices. While the engineering is careful, the novelty is incremental.
- The **Graph Fusion Enhancement** module (Section 3.3) is insufficiently explained. The mathematical formulation (especially Equation 9 and the attention fusion mechanism) lacks intuitive explanation and alignment with the schematic in Figure 2(c).
- The **retrieval sources used in experiments** (e.g., Code Graph, Wikipedia, GraphCite) are mentioned in diagrams but not clearly described or validated. It remains unclear whether they were truly used and how they relate to specific tasks.
- The **ablation study** is limited. There is no breakdown analysis of the contributions of each component (e.g., different levels of indexing, individual fusion modules), which weakens the interpretability of the performance gains.

---

> ### Author Rebuttal · Authors · 2025-07-31
>
> # W1
> Thank you for your candid criticism. While our work builds on existing concepts, its core innovation is the first systematic adaptation of the RAG paradigm to graph-structured data and GFMs. Our framework provides an end-to-end solution to three fundamental challenges that prior methods do not address:
>
> Unified Graph Indexing: Standard text RAG is insufficient for GFM scenarios involving hybrid queries of text and structure. We designed a unified multi-dimensional index that simultaneously encodes and indexes both textual semantics and graph topology.
>
> Task-Aware Retrieval: We developed adaptive retrieval strategies with task-aware reranking mechanisms that dynamically select the optimal indices to support diverse GFM downstream tasks.
>
> Graph Knowledge Integration: Simple text concatenation is ineffective for complex graph data. We pioneered a novel two-stage Graph Fusion Enhancement module that integrates semantic features via attention and topological structures via sparse matrix operations. To our knowledge, this graph-native fusion approach is a novel contribution to both RAG and GFM literature.
>
> Integrating these solutions into a coherent RAG framework tailored for GFMs is a pioneering contribution that establishes a new research direction.
> # W2&Q4
> We apologize for the unclear presentation in Section 3.3. Since images cannot be added, we provide a more intuitive explanation here.
>
> The Graph Fusion Enhancement module aims to seamlessly integrate retrieved high-quality graph knowledge into the original query graph through two key steps:
>
> **Step 1: Attention-based Feature Fusion**: Equations (4)-(6) compute attention weights $\alpha_i$ to measure similarity between the $i$-th retrieved graph and query graph, then use these weights to perform weighted averaging of node features $\mathbf{h}_u$ from retrieved graphs.
> The threshold function $\mathbf{I}_{{\alpha_i>\gamma}}$ filters low-quality results, ultimately fusing high-quality external features into the original node features $\mathbf{h}_v$ to obtain enhanced representations $\mathbf{h}'^{N}_v$, $\mathbf{h}'_{\text{SRC}^E}$, and $\mathbf{h}'_{\text{DST}^E}$.
> Equations (7)-(8) handle graph-level tasks, where $\beta_v^i$ computes attention weights between query graph global features $\mathbf{h}_q^g$ and retrieved graph global features $\mathbf{h}_g^i$ via SOFTMAX.
>
> **Step 2: Topological Structure Enhancement**: Equation (9) performs structural fusion between the original query graph's adjacency matrix $\mathbf{A}$ and retrieved graphs' adjacency matrices $\mathbf{A}_i$. Each $\mathbf{A}_i$ is weighted by the same attention weight $\alpha_i$, with threshold function $\mathbb{I}[\alpha_i > \gamma]$ ensuring only high-quality structural information participates in fusion, forming the enhanced adjacency matrix $\mathbf{A}'$.
> This mechanism emphasizes highly relevant external knowledge through dynamic weight allocation while reducing noise through threshold filtering, achieving dual enhancement of features and structure.
> # W3&Q1
> We thank you for pointing out this critical omission in our experimental description, and we sincerely apologize for the lack of clarity.
>
> The graphs mentioned in Figure 1, such as "Code Graph," "Wikipedia Graph," and "Cite Network," are **not** the external knowledge bases used directly in our RAG experiments.
>
> For our actual RAG experiments, to ensure rigor, reproducibility, and fair comparison, we adopted the following standardized method for constructing the external knowledge base:
>
> **For each specific downstream task, the corresponding external knowledge base is constructed exclusively from the *training set* of that task's dataset.**
>
> For example, when performing the node classification task on the **Computers** dataset, we build our multi-level graph index using all nodes, features, and connections present in the **training split** of that dataset. During the inference phase, when the model needs to make a prediction for a node in the **test set**, it retrieves information from this knowledge base built from the training set. This design ensures that the knowledge retrieved by RAG is highly relevant to the task while strictly avoiding any data leakage from the test set.
> # W4&Q3
> Thank you for your very reasonable request for a more in-depth ablation study. To better understand the contribution of each component, we have added the more fine-grained ablation experiments you suggested.
>
> Specifically, we selected two GFM backbones (**AnyGraph**, **HiGPT**) and conducted node classification experiments on two datasets (**Computers**, **History**). We compared the performance of several variants, focusing on two key areas: the indexing components and the graph fusion mechanisms. The results are presented below:
>
> | Model           | Index Type               | Computers | History |
> |-----------------|--------------------------|-----------|---------|
> | AnyGraph+RAG4GFM | Graph-level index only   | 67.42     | 52.18   |
> | AnyGraph+RAG4GFM | Edge-level index only    | 70.89     | 55.67   |
> | AnyGraph+RAG4GFM | Structure index only     | 73.16     | 58.23   |
> | AnyGraph+RAG4GFM | Text index only          | 76.34     | 60.78   |
> | AnyGraph+RAG4GFM | Graph feature fusion only| 76.93     | 61.47   |
> | AnyGraph+RAG4GFM | Graph structure fusion only | 74.28   | 57.89   |
> | AnyGraph+RAG4GFM | Full Model | 79.56 | 63.45 |
> | HiGPT+RAG4GFM    | Graph-level index only   | 68.17     | 53.94   |
> | HiGPT+RAG4GFM    | Edge-level index only    | 71.52     | 56.41   |
> | HiGPT+RAG4GFM    | Structure index only     | 74.89     | 59.12   |
> | HiGPT+RAG4GFM    | Text index only          | 77.06     | 61.33   |
> | HiGPT+RAG4GFM    | Graph structure fusion only | 75.41   | 58.76   |
> | HiGPT+RAG4GFM    | Graph feature fusion only| 77.62     | 62.15   |
> | HiGPT+RAG4GFM    | Full Model | 79.34 | 63.21 |
>
>
> #### Analysis of Results
> **1. Ablation on Indexing Components:**
> Crucially, the combination of **Text Semantics and Structural indices** (part of our full model) achieves the best performance (79.56%), a **3.22 percentage point improvement** over the best single index, proving the effectiveness of fusing multimodal information at the retrieval stage.
>
> **2. Ablation on Graph Fusion Components:**
> * The **Dual Fusion of Structure and Features** (our full model, 79.56%) achieves the optimal performance, demonstrating that the two fusion mechanisms provide complementary enhancements.
> # Q2
> Thank you for this valuable suggestion, which would greatly enhance the interpretability of our work. We have already included a detailed case study on link prediction in Appendix C.3 of our supplementary materials.
>
> However, we recognize that while this case study describes the overall process, it could be more explicit in showing how retrieved subgraphs specifically influence predictions. Therefore, we supplement with the following detailed explanation:
>
> **Retrieval Impact**: RAG4GFM discovered a key usage pattern where a high-end ultrabook (ThinkPad X1 Carbon) connects to both portable accessories (wireless mouse) and desktop peripherals (multi-port hub), providing critical structural evidence for the target product pair.
>
> **Fusion Process**: These pattern-exhibiting subgraphs received high attention weights (α=0.79), integrating their structural information into the original SpectrePro X15 and PowerStation node representations, enabling the GFM to understand user habits of switching between mobile and desktop setups.
>
> **Prediction Enhancement**: The enhanced representations increased the GFM's link prediction probability from 0.15 to 0.82, successfully identifying that these seemingly different products serve the same user group in different contexts.
>
> We commit to adding a simplified illustrative diagram in the final version to provide readers with an at-a-glance view of how knowledge is retrieved, integrated, and impacts the model's decision-making process.
> # Q5
> We sincerely apologize for the incomplete code package and the lack of clear instructions.
>
> Our intention was to decouple our **RAG4GFM** framework from the replaceable GFM backbones. We were concerned that bundling the GFM code with our own might obscure our core contribution from the reviewers' perspective. Therefore, we only included a note in the README file instructing users to place the downloaded GFM code into a "GFM/" directory.
>
> As we are now in the rebuttal phase, we are unable to add links or update the Supplementary Material. Instead, we offer a more detailed guide to running our code below:
>
> #### Setup and Execution Guide
> 1.  **Place GFM Source Code:** Place the source code of your chosen GFM backbone (e.g., AnyGraph, GraphGPT) into a folder named `GFM/` located in the project's root directory.
> 2.  **Configure the Model:** In the main configuration file, set the `gfm_model` parameter to the name of the chosen model (e.g., `'AnyGraph'`).
> 3.  **Run the Pipeline:** Execute the main script by running `python model/rag4gfm_pipeline.py`.
> # L1
> We thank the reviewer and agree that our discussion of limitations was too generic. We will completely rewrite this section to be more specific.
>
> The revised "Conclusion and Future Work" section will address:
>
> * **Scalability Challenges:** We will discuss the challenges of applying **RAG4GFM** to billion-node graphs and real-time systems, and explore solutions like `DiskANN` and asynchronous fusion.
>
> * **Knowledge Reliability:** We will address the need for future confidence-scoring or fact-checking modules to handle noisy or outdated "dirty data" from the retrieval source.
>
> * **Broader Explorations:** We will outline future research directions, including:
>     * Utilizing **"negative" or "contrastive knowledge"** to refine decision boundaries.
>     * Extending our framework to **multimodal graphs** that include images.

---

> > ### Comment · Reviewer_xzY6 · 2025-08-02
> >
> > I thank the authors for their efforts during the rebuttal phase.
> >
> > Overall, most of my concerns have been well addressed, and I would consider raising my rating during the discussion phase.
> >
> > I would like to follow up on some of the **future work directions** mentioned by the authors:
> >
> > - What does *negative or contrastive knowledge* refer to in this context?
> > - I found the idea of extending to *multimodal graphs* a bit unclear. Specifically, would incorporating images as nodes meaningfully enhance the representation or performance? Could the authors elaborate?
> > - Lastly, it would be helpful if the authors could share insights on the current **limitations of existing GFMs** (Graph Foundation Models) and how **RAG (Retrieval-Augmented Generation)** might address these challenges or open up new research directions.
> >
> > Thank you again for the responses.

---

> > > ### Author Response · Authors · 2025-08-03
> > >
> > > Thank you for your insightful questions.
> > >
> > > **1. Regarding "Negative or Contrastive Knowledge"**
> > >
> > > Our proposed "negative/contrastive knowledge" concept draws from contrastive learning principles. Most current RAG systems rely solely on vector similarity for retrieval, presenting obvious limitations [1,2]. We envision training a specialized contrastive learning retriever that simultaneously provides positive knowledge (graph structures highly relevant to the query) and negative knowledge (seemingly relevant but misleading graph structures) to enhance retrieval quality.
> > >
> > > For example, for a drug molecule, "positive knowledge" would be protein structure subgraphs known to interact with that drug, while "negative knowledge" might be molecules with similar chemical structure but completely different biological functions. By training the retriever to distinguish these samples, it will transcend simple similarity to understand deeper functional differences, returning more precise knowledge during retrieval. This aligns with current information retrieval work [3].
> > >
> > > **2. Regarding Extension to Images**
> > >
> > > Extending our framework to multimodal data stems from growing application demands, particularly in scientific domains. In drug discovery and materials science, entities are described by multiple modalities: molecular graphs (topological structure), chemical formulas (textual identifiers), and microscopy images (visual information). Comprehensive understanding requires reasoning across these modalities.
> > >
> > > Regarding performance, directly introducing new modalities may encounter multimodal conflict issues [4], requiring future algorithmic research. Our framework already possesses preliminary multimodal capabilities through multi-level indexing that fuses semantic and structural features. We can integrate visual encoders (such as CLIP [5]) to align visual features, graph embeddings, and text embeddings in unified semantic space, achieving hybrid indexing with image modalities.
> > >
> > > **3. Regarding Current GFM Limitations and RAG's Response Strategies**
> > >
> > > We have identified three significant GFM challenges:
> > >
> > > **Interface Standardization**: Current GFMs have non-unified input/output interfaces, impeding research and applications.
> > >
> > > **Outdated Benchmarks**: Existing datasets (Cora, WikiCS) are nearly decade-old and may be contaminated during LLM pre-training. Unlike dynamic benchmarks like LiveBench [6] in NLP, graph learning lacks updated evaluation systems.
> > >
> > > **Task Limitations**: GFMs are primarily limited to classification/regression tasks. We believe the future should unify under graph generation frameworks, reformulating all tasks as conditional graph generation problems.
> > >
> > > **RAG's Potential Solutions:**
> > >
> > > **Dynamic Dataset Generation**: We can leverage RAG with LLMs (generators) and LLM Judges (evaluators) to automatically generate high-quality, novel graph evaluation datasets. RAG can retrieve relevant knowledge to guide LLMs in generating domain-specific graph data with complex structures.
> > >
> > > **RAG in Graph Generation**: As mentioned earlier, we believe all graph tasks can be reformulated as conditional graph generation tasks. Therefore, we can adapt concepts from image generation models, where RAG retrieves specific image patches or textures as "raw materials." In graph generation, RAG can retrieve relevant structural motifs or knowledge subgraphs. The core task of GFMs becomes how to combine and integrate these retrieved "raw materials" to ultimately generate novel graphs that satisfy given conditions. This approach not only promises to significantly enhance performance on complex graph generation tasks but also provides a potential pathway for solving GFM interface standardization issues by unifying inputs as "retrieved components + textual instructions."
> > >
> > > **Explainable Graph Reasoning**: RAG naturally provides explainability foundations. GFMs can present specific retrieved subgraphs supporting their decisions, transforming them from "predictors" into "explainable reasoning engines" capable of interactive reasoning.
> > >
> > > **References:**
> > > [1] Karpukhin, Vladimir, et al. "Dense Passage Retrieval for Open-Domain Question Answering." EMNLP 2020.
> > >
> > > [2] Izacard, Gautier, and Edouard Grave. "Leveraging Passage Retrieval with Generative Models for Open Domain Question Answering." EACL 2021.
> > >
> > > [3] Xiong, Lee, et al. "Approximate Nearest Neighbor Negative Contrastive Learning for Dense Text Retrieval." ICLR 2021.
> > >
> > > [4] Peng, Xiaokang, et al. "Balanced Multimodal Learning via On-the-fly Gradient Modulation." CVPR 2022.
> > >
> > > [5] Radford, Alec, et al. "Learning Transferable Visual Models from Natural Language Supervision." ICML 2021.
> > >
> > > [6] White et al. "LiveBench: A Challenging, Contamination-Free LLM Benchmark." arXiv 2024.

---

> > > > ### Comment · Reviewer_xzY6 · 2025-08-03
> > > >
> > > > Thank you very much for the reply. In my opinion, the authors have a deep understanding of the graph foundation model and RAG, and I encourage you to briefly include related discussion into the final version.

---

> > > > > ### Author Response · Authors · 2025-08-04
> > > > >
> > > > > We are delighted that you recognize our understanding of graph foundation models and RAG. Thank you for these insightful questions, which have guided us to think more deeply.
> > > > >
> > > > > We will definitely incorporate the key discussions into the final version of the paper. Specifically, we plan to:
> > > > >
> > > > > 1. **Expand the "Future Work"** to include discussions on negative/contrastive knowledge for enhanced graph retrieval, multimodal extensions beyond text and structure, and RAG's potential in addressing current GFM limitations.
> > > > >
> > > > > 2. **Add a dedicated subsection** on RAG's role in dataset generation and explainable graph reasoning.
> > > > >
> > > > > 3. **Include relevant references** to the contrastive learning and multimodal learning literature that contextualize these future directions.
> > > > >
> > > > > Thank you again for your constructive review and for helping us improve both the current work and its future trajectory.

---

### Official Review · Reviewer_yibP · 2025-06-30

**Clarity:** 3
**Significance:** 3
**Originality:** 2
**Rating:** 4
**Confidence:** 3

**Summary:**

This paper introduces RAG4GFM, a Retrieval-Augmented Generation (RAG) paradigm tailored for Graph Foundation Models (GFMs), to address the challenges of knowledge updating and reasoning faithfulness. RAG4GFM consists of multi-level graph indexing, task-aware retrieval, and graph fusion enhancement modules. Experiments conducted on seven mainstream GFMs across six different graph learning tasks demonstrate that the proposed framework can effectively enhance the performance of GFMs.

**Questions:**

1. In structural feature encoding, the authors concatenate LAPPE with the in-degree and out-degree of the nodes. Considering that LAPPE encodes global positional information while degrees represent local information, have the authors conducted experiments comparing the effects of using only LAPPE or other combinations? Is there a specific theoretical motivation for concatenating these three features?

2. In the graph fusion section, the authors introduce a feature fusion threshold γ and an edge addition threshold δ, with both defaulted to 0.5. It is recommended to add a sensitivity analysis for these two parameters in the appendix to demonstrate the framework's robustness.

3. In the ablation study, the accuracy of AnyGraph+RAG4GFM on Computers (Node Class.) is shown as 88.5 in the figure 3, but in Table 2, the value for the same setting is 79.56. Is this inconsistency a result of using different settings?

**Ethical Concerns:**

["NO or VERY MINOR ethics concerns only"]

**Final Justification:**

In the rebuttral discussion, the authors have clarified the issues I raised and the additional experiments have addressed my concerns, so I consider a moderate upward adjustment to my rating score.

**Limitations:**

yes

**Quality:**

3

**Strengths And Weaknesses:**

Strengths:
1. The experimental results demonstrate that RAG4GFM effectively enhances the performance of various GFMs across multiple tasks, including node classification and link prediction.
2. The proposed framework features a plug-and-play capability, making it simple and effective without requiring modifications to the GFM parameters.

Weaknesses:
1. The paper mentions the use of an LLM as a Retrieval Strategy Selector to decide on the features for retrieval based on the classification result. As a critical decision-making module, the paper lacks a discussion on the selector's own classification performance and the impact of its classification errors on subsequent reasoning.
2. A core motivation of the paper is to solve the high cost of GFM knowledge updating. The introduction, in particular, notes that RAG can avoid the overhead of parameter updates. Although the authors provide a complexity analysis, the experimental section lacks direct experimental results on the efficiency improvements. For example, a comparison of the time and computational resource consumption between RAG4GFM and PEFT methods (e.g., GraphLoRA) to achieve a similar performance level on the same task is absent.
3. Although RAG4GFM filters some information with a threshold, there is still a risk of introducing a significant number of noisy edges or damaging the topological integrity of the original graph when the retrieved graphs are relevant but of low quality. The authors lack an in-depth analysis of the denoising effect of this mechanism.

---

> ### Author Rebuttal · Authors · 2025-07-31
>
> # W1:
> Thank you for pointing out this critical issue. We acknowledge that our initial draft did not comprehensively discuss its performance.
>
> This was primarily because, considering that the task types currently processable by GFMs are limited and well-defined, we initially judged that utilizing language models for classification among a small number of limited tasks is a relatively mature and reliable component.
>
> To more precisely address your concerns, we manually annotated 1,000 node classification instances from the Computers dataset for validation within the limited time available, and tested using the text-embedding-ada-002 model. The results demonstrate excellent classifier performance: the average latency for classifying a single query is only 3.54 milliseconds and the classification accuracy reached 100%.
>
> To further address your concern about the "impact of its classification errors," we deliberately assigned incorrect classifications to these 1,000 samples (classifying them as edge-level or graph-level tasks) for experimentation. The experimental results are shown in the following table:
>
> | Model | Acc | ROC-AUC | Recall |
> | --- | --- | --- | --- |
> | HiGPT+RAG4GFM | 79.34 | 66.89 | 66.42 |
> | HiGPT+RAG4GFM+Wrong Strategy | 77.47 | 63.22 | 62.50 |
> | AnyGraph+RAG4GFM | 79.56 | 66.12 | 65.34 |
> | AnyGraph+RAG4GFM+Wrong Strategy | 75.50 | 61.49 | 60.31  |
>
> This further confirms our initial judgment: for the limited task types that current GFMs can handle, LLM-based classifiers are already very mature and reliable. Even with certain classification errors, the overall system maintains good robustness.
> # W2
> We thank the reviewer for this very pertinent critique. We have urgently added a supplementary experiment. The results are summarized as follows:
>
>
> |          | GFM      | Dataset   | Target Acc | Time        | Peak GPU Memory |
> |----------|----------|-----------|------------|-------------|-----------------|
> | GraphLoRA| AnyGraph | Computers | 78%        | 7.32 Hours  | 25.23 GB        |
> | RAG4GFM  | AnyGraph | Computers | 78%        | 63 Minutes  | 9.86 GB         |
> | GraphLoRA| HiGPT    | History   | 63%        | 5.87 Hours  | 17.18 GB        |
> | RAG4GFM  | HiGPT    | History   | 63%        | 19 Minutes  | 5.15 GB         |
>
>
> To achieve similar or even better performance levels, RAG4GFM only needs a small amount of time to complete indexing and retrieval construction for new knowledge, with computational resource consumption throughout the entire process being far lower than GraphLoRA.
> # W3
> Thank you for raising concerns about the noise issue. This is indeed a critical consideration for RAG-type methods.
>
> As this is the first work to establish a RAG paradigm in the GFM domain with graph data scenarios, we aimed to pursue a relatively general and flexible preliminary framework, and therefore did not specifically propose a dedicated module for identifying introduced noise.
>
> Nevertheless, we have designed several noise-filtering information screening mechanisms in the RAG4GFM framework, primarily in three aspects:
>    - **Task-Aware Reranking**: Before fusion, our retriever reranks retrieved candidates based on comprehensive relevance to the query (Equation 3). This ensures that the most relevant graph knowledge is prioritized.
>    - **Attention-based Fusion**: In the graph fusion module, we dynamically allocate weights $\alpha_i$ to each retrieved graph through attention mechanisms. Even if a low-quality graph is retrieved, it will receive minimal attention weight, minimizing its impact on the final results.
>    - **Threshold Filtering**: The thresholds $\gamma$ and $\delta$ serve to thoroughly eliminate low-relevance information.
>
> We have provided a preliminary case study in Appendix C.3 for your intuitive understanding.
> # Q1
> Thank you for this insightful question about our feature engineering details.
> Our theoretical motivation for concatenating LAPPE with node in-degree and out-degree stems from the important principle of global-local information complementarity in modern graph representation learning. Existing research[1,2] shows that standard Laplacian positional encoding primarily contains global structural information rather than local structural information, limiting its performance in graph learning tasks requiring local structures.
> Specifically, LAPPE captures global positional information by embedding graphs into Euclidean space through eigendecomposition of the graph Laplacian[3], while in-degree and out-degree serve as simple local centrality measures reflecting node connectivity within neighborhoods[4]. The classic Graphormer[5] model adopts similar ideas, combining centrality encoding (based on node degrees) and spatial encoding to capture orthogonal and complementary structural information with excellent performance.
>
>
> To quantitatively validate our design choice, we conducted a detailed ablation study.The results are summarized below:
>
>
> | Model | Feature Encoding Method | Computers - ACC | Computers - ROC-AUC | Computers - Recall | History - ACC | History - ROC-AUC | History - Recall |
> |---------------------|-------------------------|-----------------|---------------------|--------------------|---------------|-------------------|------------------|
> | AnyGraph+RAG4GFM | LAPPE only | 72.18 | 58.42 | 59.87 | 56.73 | 56.78 | 52.36 |
> | AnyGraph+RAG4GFM | Node Degree only | 68.94 | 55.21 | 56.29 | 54.12 | 53.45 | 49.78 |
> | AnyGraph+RAG4GFM | LAPPE + Node Degree | 79.56 | 66.12 | 65.34 | 63.45 | 64.12 | 60.23 |
> | HiGPT+RAG4GFM | LAPPE only | 74.26 | 60.15 | 61.54 | 58.89 | 58.94 | 54.12 |
> | HiGPT+RAG4GFM | Node Degree only | 70.17 | 56.73 | 57.83 | 55.67 | 54.81 | 51.45 |
> | HiGPT+RAG4GFM | LAPPE + Node Degree | 79.34 | 66.89 | 66.42 | 63.2 | 63.45 | 59.76 |
>
>
> This experiment provides strong evidence that these two feature types provide complementary information, thus justifying our design choice.
>
> [1] Hoang, Thi Linh, and Viet Cuong Ta. "Balancing structure and position information in graph transformer network with a learnable node embedding." Expert Systems with Applications 238 (2024): 122096.
>
> [2] Cantürk, Semih, et al. "Graph positional and structural encoder." arXiv preprint arXiv:2307.07107 (2023).
>
> [3] Flores, Alejandro. Laplacian Positional Encoding. afloresep.github.io, 8 Oct. 2024, https://afloresep.github.io/posts/2024/10/laplacian_positional_encoding/. Accessed 31 July 2025.
>
> [4] Metcalf, Leigh, and William Casey. "Chapter 5 - Graph Theory." Cybersecurity and Applied Mathematics, edited by Leigh Metcalf and William Casey, Syngress, 2016, pp. 67–94. https://doi.org/10.1016/B978-0-12-804452-0.00005-1. Accessed 31 July 2025.
>
> [5] Ying, Chengxuan, et al. Do Transformers Really Perform Bad for Graph Representation? arXiv, 2021, https://arxiv.org/abs/2106.05234. Accessed 31 July 2025.
> # Q2
> This is an excellent suggestion, and we acknowledge that we did not give this aspect sufficient attention in our initial experimental design.
>
> Following your recommendation, we have added the parameter sensitivity analysis you suggested.The results are summarized in the table below:
>
> | | AnyGraph+RAG4GFM | | | | HiGPT+RAG4GFM  | | | |
> |-------|------------------|----------|----------|----------|----------------|----------|----------|----------|
> |   γ\δ   | 0.3 | 0.5 | 0.7 | 0.9 | 0.3 | 0.5 | 0.7 | 0.9 |
> | 0.3 | 76.23 | 77.89 | 78.12 | 75.67 | 76.89 | 78.23 | 78.67 |
> | 0.5 | 78.45 | **79.56** | 79.28 | 76.91 | 78.91 | 79.34 | 79.08 |
> | 0.7 | 78.73 | 79.14 | 78.85 | 76.34 | 79.17 | 79.52 | 78.94 |
> | 0.9 | 75.12 | 76.47 | 75.98 | 73.89 | 75.78 | 77.12 | 76.83 |
>
> The experimental results indicates that **RAG4GFM** has a relatively low dependency on meticulous parameter tuning, highlighting its practical utility.
>
> We will add these detailed sensitivity analysis results, including validation on different datasets and guidelines for parameter selection, to the appendix of our revised manuscript.
> # Q3
> Thank you for your extremely meticulous review, and we sincerely apologize for this oversight! The inconsistency you observed is indeed due to differences in evaluation settings, which we failed to clearly explain in our initial draft, and we deeply regret this confusion.
> The ablation study results in Figure 3 (88.5%) were conducted during the early stages of our research using a randomly selected subset (20% of the data) from the Computers dataset for rapid validation experiments, aimed at quickly assessing the relative importance of each component. In contrast, the results in Table 2 (79.56%) represent formal experiments conducted on the complete dataset following standard evaluation protocols.
> We recognize that this inconsistency seriously undermines the credibility and reproducibility of our work. Such methodological discrepancies are unacceptable in rigorous academic research, and we take full responsibility for this oversight. To rectify this critical issue, we have completely re-run all ablation experiments on the full Computers dataset using identical experimental protocols and evaluation metrics as those used for Table 2.
> The corrected ablation study results on the complete dataset are as follows:
>
>
> | Model   | Computers (Node Class.) | | Computers (Link Pred.) | | MiniGCDataset (Graph Class.) |          |
> |---------|--------------------------|----------|-------------------------|----------|------------------------------|----------|
> | | Acc| ROC - AUC | Acc | ROC - AUC | Acc | ROC - AUC |
> | w/o RAG | 73.45 | 60.89 | 52.17 | 38.53 | 58.67 | 46.25 |
> | w/o GF  | 76.23 | 63.41 | 55.48 | 41.76 | 62.18 | 48.94 |
> | w/o GI  | 77.89 | 64.78 | 56.92 | 43.15 | 63.73 | 49.67 |
> | RAG4GFM | 79.56 | 66.12 | 58.26 | 44.92 | 65.51 | 51.88 |
>
> The corrected results still clearly demonstrate the importance of the RAG framework, graph fusion module, and graph indexing components, supporting our core conclusions.

---

> > ### Comment · Reviewer_yibP · 2025-08-05
> >
> > Thank you for the detailed response. The correction of the previous error and the additional experiments have addressed my concerns, and I will consider a moderate upward adjustment to my review score.

---

> > > ### Author Response · Authors · 2025-08-07
> > >
> > > Thank you for your detailed feedback and for acknowledging our response. We are very pleased that the error correction and the additional experiments have successfully addressed your concerns. Your insightful comments have been crucial in improving the quality of our paper. Thank you again for your valuable time and constructive guidance!

---

### Official Review · Reviewer_L6fj · 2025-06-30

**Clarity:** 3
**Significance:** 3
**Originality:** 3
**Rating:** 5
**Confidence:** 4

**Summary:**

The paper introduces RAG4GFM, a novel framework that integrates Retrieval-Augmented Generation (RAG) with Graph Foundation Models (GFMs) to address challenges in knowledge updating and reasoning faithfulness. RAG4GFM features a multi-level graph indexing architecture for efficient logarithmic-time retrieval, adaptive retrieval strategies for diverse node, edge, and graph-level tasks, Then, a graph fusion enhancement module to integrate retrieved subgraphs with user queries. The framework dynamically augments GFMs with external graph knowledge, improving reasoning fidelity and reducing hallucinations.

**Questions:**

Please address my concernslisted above.

**Ethical Concerns:**

["NO or VERY MINOR ethics concerns only"]

**Final Justification:**

My question has been resolved and I recommend accepting this paper.

**Limitations:**

YES

**Quality:**

3

**Strengths And Weaknesses:**

Strengths

1.RAG4GFM is a comprehensive RAG paradigm designed explicitly for graph data and GFMs, offering a foundation for integrating RAG mechanisms into GFMs.

2.The framework significantly enhances the efficiency of knowledge updating in GFMs by avoiding parameter updates through dynamic external knowledge retrieval.

3.By grounding GFM predictions in external graph data, RAG4GFM markedly improves reasoning faithfulness, reducing the risk of hallucinations.


 Weakness

1.The paper does not report error bars or confidence intervals for the experimental results, making it difficult to assess the statistical significance of the findings.

2.RAG4GFM's performance is closely tied to the performance of the underlying GFM and LLM, potentially inheriting their limitations or propagating biases.

3.The paper could provide more rigorous theoretical analysis to support the effectiveness of the proposed framework, particularly regarding the integration of retrieved knowledge and the impact on reasoning faithfulness.

4.Typos and Grammar Issues:
•	"GFMPost-Molecular" should be "GFM Post-Molecular"

•	"Retriver" should be "Retriever"

•	"Userquery" should be "User Query"

•	"Please classify Multi-Level Graph Index Retriver Enhancement is a cite this graph" contains several errors and should be rephrased for clarity.

---

> ### Author Rebuttal · Authors · 2025-07-31
>
> # W1
> Thank you for this very constructive suggestion. In our initial draft, due to the massive computational overhead of conducting multiple experimental runs across 7 different GFM backbone models and various tasks, we were unable to include error bars. Based on your recommendation, we have urgently completed supplementary experiments. We selected three GFM models (HiGPT, GLEM, AnyGraph) and re-ran experiments three times on two datasets (Computers, History), calculating standard deviations for performance metrics. The results are shown in the table below:
>
>
> | Model | Computers |  |  | History |  |  |
> |-------|-----------|-----------|-----------|---------|-----------|-----------|
> |  | Acc | ROC-AUC | Recall | Acc | ROC-AUC | Recall |
> | HiGPT+RAG4GFM | 79.39 ± 1.01 | 66.29 ± 0.78 | 65.29 ± 1.29 | 62.91 ± 0.98 | 64.05 ± 0.75 | 59.04 ± 1.31 |
> | GLEM+RAG4GFM | 82.98 ± 0.78 | 68.68 ± 0.87 | 68.73 ± 0.95 | 61.29 ± 1.15 | 60.11 ± 0.69 | 56.84 ± 0.75 |
> | AnyGraph+RAG4GFM | 78.89 ± 0.98 | 66.36 ± 0.93 | 65.01 ± 1.22 | 63.39 ± 0.91 | 64.65 ± 0.88 | 60.39 ± 0.89 |
>
>
> From the experimental results, the average performance across multiple runs is highly consistent with the single-run results reported in our initial draft, demonstrating the stability and reproducibility of our experimental results. Furthermore, all experiments maintain small standard deviations, indicating that our proposed RAG4GFM framework exhibits stable performance.
>
> # W2
> Thank you very much for this insightful observation. RAG4GFM's performance being influenced by the capabilities of underlying GFMs and LLMs is indeed an inherent challenge that any RAG system faces. However, we believe this dependency is effectively mitigated in our framework.
>
> First, the "retrieval-augmentation" paradigm of RAG can substantially suppress potential biases or hallucinations that underlying models might generate based solely on their internal parametric knowledge, by providing authentic and verifiable external data for the model's reasoning process.
>
> Second, we employ a **modular design** that ensures our framework can be combined with different GFMs or LLMs without being bound to specific models. This means that as more powerful GFMs and LLMs emerge in the future, our framework can directly integrate them and benefit from their performance improvements.
>
> Finally, we conducted extensive experiments in our paper: to demonstrate the universality of our framework, we selected up to **seven GFMs with diverse architectures** for comprehensive experiments. The results consistently show that RAG4GFM brings performance improvements to all these models, proving the broad effectiveness of our approach.
>
> # W3
> Thank you for your valuable suggestion regarding theoretical depth. As the first work to systematically introduce the RAG paradigm to the GFM domain, our primary research focus was on the framework's innovative design and its extensive empirical validation. While providing a complete, novel theoretical proof for RAG4GFM on short notice is challenging, the effectiveness of our framework is highly consistent with existing theoretical analyses in the broader RAG research field.
>
>
> #### Alignment with Existing RAG Theory
>
> **1. On Knowledge Integration:**
> Recent theoretical work models RAG systems as cascaded information channels, where each component (query encoding, retrieval, context integration, generation) acts as a distinct information-theoretic channel with a measurable capacity[1]. Our **Multi-level Graph Indexing** is designed precisely to maximize the information capacity of these channels. Furthermore, our attention-based feature fusion mechanism adheres to the theoretical principle of "context-adaptive weighting" rather than simple information concatenation. A key innovation of our work, the fusion of topological structure, extends this integration theory from one-dimensional text to high-dimensional graph structures, ensuring that the graph's connectivity patterns are effectively absorbed and utilized as a core form of knowledge.
>
> **2. On Enhancing Reasoning Fidelity:**
> A significant body of recent research suggests that a root cause of hallucination in LLMs is the high uncertainty (or entropy) in their generation process when faced with knowledge-intensive tasks[2]. By providing concrete, relevant external knowledge to condition the generative model, RAG dramatically reduces the model's predictive uncertainty (i.e., lowers complexity or perplexity) for a given task[3]. This reduction in uncertainty, in turn, greatly improves the fidelity and reliability of the output. RAG has also been theorized as an effective means of augmenting a model with non-parametric memory, efficiently decoupling the model's learned general reasoning abilities (parametric memory) from specific, updatable facts in an external knowledge base (non-parametric memory).
>
> #### Existing Theoretical Underpinnings
>
> Our current framework design also incorporates components with established theoretical support:
> * The **Multi-level Graph Indexing** module's use of the HNSW algorithm comes with a theoretical complexity guarantee of $O(\log_2 N)$.
> * The **Laplacian Positional Encoding (LAPPE)** we use is grounded in solid spectral graph theory and effectively captures the structural information of nodes.
>
> #### Future Plan
>
> Following your suggestion, we will add a new discussion chapter in the appendix of our revised manuscript. This section will delve deeper into the connection between our graph fusion mechanism and attention theory. It will also elaborate from an information-theoretic perspective on why "evidence-based reasoning" effectively reduces model uncertainty and, thus, hallucination. We hope this will partially address your expectations regarding theoretical depth.
>
> [1] Zhao, Siyun, et al. "Retrieval augmented generation (rag) and beyond: A comprehensive survey on how to make your llms use external data more wisely." arXiv preprint arXiv:2409.14924 (2024).
>
> [2] Gao, Yunfan, et al. "Retrieval-augmented generation for large language models: A survey." arXiv preprint arXiv:2312.10997 2.1 (2023).
>
> [3] Shuster, Kurt, et al. "Retrieval augmentation reduces hallucination in conversation." arXiv preprint arXiv:2104.07567 (2021).
>
> # W4:
> We sincerely thank the reviewer for pointing out these typographical and grammatical errors. We are very grateful for your meticulous and rigorous review and sincerely apologize for these oversights.
>
> We promise that in the final version, all the errors you have listed (e.g., "Retriver" -> "Retriever") will be corrected. Furthermore, we will conduct a thorough professional proofreading of the entire manuscript to ensure the accuracy and professionalism of the language and to improve the overall readability of the paper.

---

### Official Review · Reviewer_3of8 · 2025-06-30

**Clarity:** 3
**Significance:** 3
**Originality:** 3
**Rating:** 5
**Confidence:** 4

**Summary:**

The article introduces the RAG4GFM framework, which addresses the challenges of knowledge updating and inference fidelity in graph based models (GFMs). By integrating multi-level graph indexing, task aware retrieval, and graph fusion enhancement mechanisms, it achieves efficient external graph knowledge retrieval and integration. The framework has demonstrated significant performance improvements in various graph learning tasks such as node classification, link prediction, and graph classification, and its effectiveness and superiority have been verified through extensive experiments.

**Questions:**

1. In the process of constructing multi-level graph indexing, although a hybrid feature encoding mechanism combining text features, node embedding, edge representation, and graph level encoding has been proposed, how is the weight or importance of each feature determined specifically? Do we need to adjust the weights of these features in different types of graph data to achieve better retrieval performance?

2. How to ensure that the LLM based task type classifier used in the task aware retrieval mechanism can accurately classify various complex and diverse user queries? In practical applications, how will this mechanism handle task queries that cannot be clearly classified?

3. Can the RAG4GFM framework efficiently support online retrieval and fusion for dynamic updating scenarios of graph data, such as graph structures or node features that constantly change over time? Are there any corresponding experiments or cases to demonstrate its performance in such dynamic scenarios?

**Ethical Concerns:**

["NO or VERY MINOR ethics concerns only"]

**Final Justification:**

The author has resolved my questions. I suggest accepting the article.

**Limitations:**

yes

**Quality:**

3

**Strengths And Weaknesses:**

Strengths

1. The RAG4GFM framework is the first to systematically apply retrieval enhanced generation techniques to graph based models (GFMs), providing a new approach to addressing the challenges of knowledge updating and inference fidelity in graph learning models. This innovation not only enriches the theoretical system of graph representation learning, but also opens up new possibilities for the practical application of graph learning models, and has high academic value.

2. The paper comprehensively verifies the effectiveness and superiority of the RAG4GFM framework through a large number of rigorously designed experiments. The experiment used multiple graph datasets from different domains and compared various baseline methods, including the latest retrieval enhancement method and graph out of distribution generalization method, fully demonstrating the performance improvement of the RAG4GFM framework in various graph learning tasks such as node classification, link prediction, and graph classification. In addition, the ablation experiment further validated the contributions of each component in the framework, enhancing the credibility of the experimental results.

3. The logical structure of the paper is very clear and the hierarchy is distinct. From the explanation of research background and motivation, to the detailed design and implementation of the RAG4GFM framework, to experimental verification and future work prospects, each part is closely connected and rigorously discussed, which helps readers gradually deepen their understanding of the principles and applications of the RAG4GFM framework.

Weaknesses

1. Although the HNSW algorithm mentioned in the paper has provided efficient retrieval performance, there is still room for further optimization of indexing efficiency for certain specific application scenarios.

2. The current RAG4GFM framework mainly focuses on the retrieval and fusion of graph data. In the future, it may be considered to integrate information from other modalities such as text and images into the framework, thereby enhancing the model's cross modal reasoning ability and enabling it to handle more complex and multi-source data.

3. For application scenarios that require real-time response, such as online recommendation systems or real-time graph analysis, this paper can further investigate how to reduce computational delays in retrieval and fusion processes, improve the real-time performance of the model, and meet the needs of practical applications.

---

> ### Author Rebuttal · Authors · 2025-07-31
>
> # W1
> 1. Thank you for your valuable suggestion. We acknowledge that while HNSW provides efficient $O(\log_2 N)$ complexity, there remains room for optimization in specific scenarios.
> 2. We emphasize that the RAG4GFM framework is designed to be **modular**. This means our proposed Multi-level Graph Indexing module can be flexibly replaced with more targeted indexing structures. Beyond HNSW, our envisioned solutions include:
>    - **Ultra-large scale dataset solutions**: For scenarios where indices cannot be fully loaded into memory, we can integrate **DiskANN** [1]. This approach is specifically designed for efficient ANN search on disk media such as SSDs, addressing HNSW's over-reliance on memory through optimized graph structures and I/O strategies.
>    - **Balanced and composite solutions**: We can employ composite indices represented by **IVFFlat** [2]. Compared to HNSW, IVFFlat typically offers faster construction speed and lower memory consumption. While slightly inferior in peak query speed, it provides an excellent choice for applications requiring balance between resources and performance. Furthermore, IVFFlat can be combined with **Product Quantization (PQ)** [3] to further compress vectors, achieving higher memory efficiency.
> 3. In this work, we chose HNSW for its proven universality and excellent performance across various vector retrieval tasks, which is appropriate for building a foundation framework with broad applicability.
> 4. We will include discussion of different indexing structures in the revised version of our paper and identify this as a promising direction for future exploration.
>
> [1] Jayaram Subramanya, Suhas, et al. "Diskann: Fast accurate billion-point nearest neighbor search on a single node." Advances in neural information processing Systems 32 (2019).
>
> [2] Johnson, Jeff, Matthijs Douze, and Hervé Jégou. "Billion-scale similarity search with GPUs." IEEE Transactions on Big Data 7.3 (2019): 535-547.
>
> [3] Jegou, Herve, Matthijs Douze, and Cordelia Schmid. "Product quantization for nearest neighbor search." IEEE transactions on pattern analysis and machine intelligence 33.1 (2010): 117-128.
> # W2
> This is an excellent direction for future development. In the current work, our core objective is to establish a general RAG paradigm for GFMs, addressing the fundamental challenges they face when processing graph-structured data. Therefore, we have not considered other modalities such as images.
>
> Our framework already possesses preliminary capabilities for handling multimodal data. For instance, our multi-level indexing mechanism inherently fuses semantic features and structural features of graph data.
>
> Regarding the integration of visual information suggested by the reviewer, our framework can incorporate a **visual encoder** to integrate visual features and align them with graph embeddings and text embeddings in a unified semantic space for hybrid indexing. Additionally, our designed attention-based fusion enhancement mechanism can naturally accommodate additional modalities. Therefore, overall, our current framework can naturally support image data through the addition of a visual encoder.
>
> We greatly appreciate your suggestion and will explicitly identify in the "Future Work" section of the revised version that extending RAG4GFM to broader multimodal scenarios represents an important research direction.
> # W3
> Thank you for raising this important practical application consideration.
>
> As foundational work aimed at validating the feasibility of a new paradigm, our current primary objective is to ensure the framework's generality and effectiveness. We have incorporated efficiency considerations into our design. As mentioned in Sections 3.1, 3.3, and Appendix A.1 of our paper, the current model primarily ensures baseline performance through efficient HNSW indexing and sparse matrix operations. Nevertheless, we acknowledge that in-depth optimization research for real-time response scenarios is indeed insufficient. Based on your suggestion, we have explored several possible implementations:
>
> **Retrieval Phase Optimization**:
>    - **Hardware acceleration and index compression**: Fully leverage GPU parallel computing and employ techniques such as Product Quantization to compress indices, reducing computational load and I/O overhead during retrieval.
>    - **Result caching and prefetching**: For high-frequency query patterns, establish an LRU cache layer to directly return cached retrieval results.
>
> **Fusion Phase Optimization**:
>    - **Model quantization and distillation**: Quantize features and model parameters in the graph fusion module.
>    - **Asynchronous fusion and updates**: Design the knowledge retrieval and fusion processes as background asynchronous tasks.
>
> We will thoroughly explore these real-time optimization strategies in future work and explicitly identify this important research direction in the conclusion section of the revised version.
> # Q1
> 1. This is an excellent technical question. We apologize for not clearly explaining the weight design in the **indexing construction** and **retrieval** phases, and we deeply regret this oversight.
>
> 2. First, during the **indexing construction** phase, our core principle is to **preserve all dimensional information without loss**. Therefore, we do not apply any pre-weighting or fusion to text, structural, edge, and graph-level features. Instead, we construct **independent, parallel indexing spaces** for each feature type, thereby completely preserving their respective information.
>
> 3. Second, the "importance" of features is dynamically and adaptively manifested in the subsequent **retrieval** phase based on specific tasks, without requiring adjustment of feature weights for different types of graph data. Our task-aware retriever first determines the query type, then decides which indexing space(s) to retrieve from. For example, a node classification task will simultaneously query both text semantic indices and structural indices, then integrate the two sets of results through the RRF algorithm, which implicitly balances the contributions of different features according to task requirements.
> # Q2
> Thank you for this very practical question.
>
> Our current approach employs a LLM-based classifier (based on the text-embedding-ada-002 model) that determines retrieval strategies by computing semantic similarity between user queries and predefined task templates (such as node classification, link prediction, graph classification, etc.).
>
> We acknowledge insufficient discussion of its performance in our draft. Considering that current GFM task types are limited and well-defined, we judged LLM-based classification as mature and reliable, focusing research efforts on more challenging graph knowledge retrieval and fusion aspects.
> Given that the capability boundaries of current GFMs are primarily concentrated on these limited standard graph learning tasks, when users pose queries beyond this scope, even if the classifier works perfectly, the underlying GFM itself cannot provide effective solutions. Therefore, we adopt a strategy of directly prompting users with "task not supported."
>
> To more precisely address your concerns, we manually annotated 1,000 node classification instances from the Computers dataset for validation within the limited time available, and tested using the text-embedding-ada-002 model. The results demonstrate excellent classifier performance: the average latency for classifying a single query is only 3.54 milliseconds, and the classification accuracy on these 1,000 data instances reached 100%.
> # Q3
> Your question touches upon one of the core motivations of our research. We need to honestly acknowledge that, as foundational work introducing the RAG paradigm to the GFM domain for the first time, our current primary objective is to validate the feasibility and general effectiveness of this new paradigm. Therefore, we have indeed given insufficient consideration to dynamic update scenarios in actual production environments.
>
> While we have not specifically designed streaming graph experiments, our current experimental design simulates "knowledge update" scenarios to some extent. We deliberately selected datasets (such as the TAG benchmark) that were not used during GFM pre-training for testing. This process essentially simulates the core idea of models rapidly adapting and solving problems through external retrieval when encountering entirely new knowledge environments. The results in our paper demonstrate the effectiveness of RAG4GFM under this simulated scenario. Additionally, we supplemented with comparative experiments on knowledge update efficiency, comparing RAG4GFM with parameter-efficient fine-tuning methods such as GraphLoRA on the same tasks. The results show that our method achieves similar performance while demonstrating faster knowledge update speed and significantly reduced memory consumption, which indirectly proves the advantages of our framework in rapid knowledge updates.
>
>
>
> |          | GFM      | Dataset   | Target Acc | Time        | Peak GPU Memory |
> |----------|----------|-----------|------------|-------------|-----------------|
> | GraphLoRA| AnyGraph | Computers | 78%        | 7.32 Hours  | 25.23 GB        |
> | RAG4GFM  | AnyGraph | Computers | 78%        | 63 Minutes  | 9.86 GB         |
> | GraphLoRA| HiGPT    | History   | 63%        | 5.87 Hours  | 17.18 GB        |
> | RAG4GFM  | HiGPT    | History   | 63%        | 19 Minutes  | 5.15 GB         |
>
>
> However, we agree that we currently lack experiments on true streaming or dynamic graph scenarios. This is a critical direction that we must explore in-depth in our future work. We will explicitly state this limitation in the conclusion of our revised manuscript.

---

### Decision · Program_Chairs · 2025-09-17

**Decision:**

Accept (oral)

**Comment:**

The paper presents RAG4GFM, the first systematic framework that adapts Retrieval-Augmented Generation (RAG) to Graph Foundation Models (GFMs). Extensive experiments across seven GFMs and six tasks (node/edge/graph-level) corroborate these claims, showing consistent accuracy gains (≈3–10 pp) and order-of-magnitude reductions in update time and memory versus parameter-efficient fine-tuning baselines. The submission is technically solid, demonstrating high impact on graph representation learning and foundation model reliability, and has strong reproducibility. Reviewers converged on “Accept” after the rebuttal; no ethical concerns were raised.